# Spatial patterns of Holocene temperature changes over mid-latitude Eurasia

Jiawei Jiang [1], Bowen Meng[1,9], Huanye Wang [2], Hu Liu[2], Mu Song[1], Yuxin He[3], Cheng Zhao[4], Jun Cheng [5], Guoqiang Chu [6] ✉, Sergey Krivonogov [7] ✉, Weiguo Liu[2] & Zhonghui Liu [1,8] ✉

The Holocene temperature conundrum, the discrepancy between proxy-based Holocene global cooling and simulated global annual warming trends, remains controversial. Meanwhile, reconstructions and simulations show inconsistent spatial patterns of terrestrial temperature changes. Here we report Holocene alkenone records to address spatial patterns over mid-latitude Eurasia. In contrast with long-term cooling trends in warm season temperatures in northeastern China, records from southwestern Siberia are characterized by colder conditions before ~6,000 years ago, thus long-term warming trends. Together with existing records from surrounding regions, we infer that colder airmass might have prevailed in the interior of mid-latitude Eurasian continent during the early to mid-Holocene, perhaps associated with atmospheric response to remnant ice sheets. Our results challenge the proposed seasonality bias in proxies and modeled spatial patterns in study region, highlighting that spatial patterns of Holocene temperature changes should be reconsidered in record integrations and model simulations, with important implications for terrestrial hydroclimate changes.

Temperature changes during the Holocene epoch provide key insights into climate projections under warming scenarios. Global syntheses of proxy-based Holocene temperature records indicate a thermal maximum during the early to mid-Holocene, followed by a cooling trend toward the preindustrial period, coincident with declining boreal summer insolation[1,2]. However, transient climate model simulations indicate a long-term Holocene warming trend in global annual mean temperature, suggested to be forced by retreating Northern Hemisphere ice sheets and rising greenhouse gas concentrations[3,4]. This model-data discrepancy, known as the Holocene temperature conundrum[3], is proposed to be associated with potential seasonal or spatial biases in proxy-based reconstructions[4–6] or sensitivity of climate models[7–9]. Recent seasonal temperature reconstructions from East Asia and the Northern Hemisphere reveal an early to mid-Holocene thermal maximum in both summer and winter seasons[10,11], not supporting the generally annual warming trends presented in model simulations or the proposed seasonality effects.

Spatial patterns of Holocene temperature changes provide important insights into dynamical responses to climate forcings and offer a regional perspective on current global warming. Yet, recent Holocene temperature syntheses and simulations have yielded inconsistent spatial patterns over the mid-latitude

[1]Department of Earth Sciences, The University of Hong Kong, Hong Kong, China. [2]State Key Laboratory of Loess and Quaternary Geology, Institute of Earth Environment, Center for Excellence in Quaternary Science and Global Change, Chinese Academy of Sciences, Xi'an 710061, China. [3]Key Laboratory of Geoscience Big Data and Deep Resource of Zhejiang Province, School of Earth Sciences, Zhejiang University, Hangzhou 310027, China. [4]School of Geography and Ocean Science, Nanjing University, Nanjing 210023, China. [5]Key Laboratory of Meteorological Disaster, Nanjing University of Information Science and Technology, Nanjing 210044, China. [6]Institute of Geology and Geophysics, Chinese Academy of Sciences, Beijing 100029, China. [7]Faculty of Geosciences and Environmental Engineering, Southwest Jiaotong University, Chengdu 610031, China. [8]Institute of Climate and Carbon Neutrality, The University of Hong Kong, Hong Kong, China. [9]Present address: Research Institute of Petroleum Exploration and Development, PetroChina, Beijing, China.
✉e-mail: chuguoqiang@mail.igcas.ac.cn; carpos.sergey@gmail.com; zhliu@hku.hk

Eurasian continent[9,11–13]. A spatiotemporal analysis of annual temperature variability using data from model simulations suggests Holocene cooling trends over the Arctic Ocean and mid-to-high latitude Eurasia[12]. A data assimilation reconstruction combining proxy records and simulation results also yields high annual mean temperatures across continental Eurasia at 6000 a BP relative to the preindustrial period and, thus, a cooling trend in annual mean temperatures since the mid-Holocene[9]. Yet, a synthesis of pollen records indicates warm early to mid-Holocene conditions followed by generally annual cooling trends in Europe and 0–50°N of Asia, but long-term Holocene annual warming trends in 50–75°N of Asia[11]. Another recent pollen-based synthesis study shows Holocene annual warming trends in 40–50°N of Asia but annual cooling trends in 60–70°N of Asia[13]. The latitudinal patterns of Holocene temperature change over Eurasia inferred from pollen records[11,13] appear to differ from the spatially unanimous annual cooling trends over the extratropical Eurasian continent in transient model simulations[9,12]. Given the uncertainties in regional temperature changes and poorly constrained climate responses to various forcings, our understanding of Holocene hydroclimatic changes and associated mechanisms also remains largely speculative[14,15].

To advance our understanding of Holocene terrestrial temperature changes and their spatial patterns, here we present alkenone records from Lake Yihesariwusu (48.13°N, 118.68°E) in northeastern China, Lake Ebeyty (54.65°N, 71.73°E), Kuchuk (52.69°N, 79.84°E), and Maloye Yarovoye (53.03°N, 79.11°E) in southwestern Siberia (Methods, Fig. 1, Supplementary Figs. 1 and 2). Our results, together with temperature records from surrounding regions, allow us to identify spatial patterns of Holocene temperature changes over the mid-latitude Eurasian continent, which sheds new light on the Holocene temperature conundrum and associated terrestrial hydrological changes.

## Results and discussion

### Alkenone proxy interpretation

We use the alkenone unsaturation index ($U_{37}^{K'}$) (Methods) as a terrestrial warm season temperature indicator. Long-chain alkenones produced by haptophytes are increasingly utilized as temperature and salinity indicators in saline lakes and marine environments. The $U_{37}^{K'}$-temperature calibration from mid-latitude Asian lakes[16] is largely consistent with a culture experiment[17], and the $U_{37}^{K'}$ index has been successfully applied to reconstruct Holocene temperature changes in this region[18–20]. Phylogenetic studies[21,22] suggest that Group I haptophytes, with the typical presence of $C_{37:3}$ isomer, tend to live in freshwater lakes, while Group II haptophytes live in brackish/saline lakes. Phylogenetic groups may have different temperature sensitivities[23]. However, alkenone $C_{37}$ isomer has not been identified from all four lakes in our study, indicating that alkenone-based records in our study are not affected by potential phylogenetic effects during the Holocene. The alkenone $U_{37}^{K}$ index, which includes $C_{37:4}$ in its definition, has been used to reconstruct temperature changes in high latitude lakes[24,25], where $C_{37:4}$ abundance appears to primarily respond to temperature variations. Nevertheless, alkenone $C_{37:4}$ responds more sensitively to lake salinity changes in mid-latitude arid/semi-arid regions where lake salinity could vary substantially[26–28], and the proportion of alkenone $C_{37:4}$ index (%$C_{37:4}$, Methods) has been successfully applied to reconstruct lake salinity changes in northwestern China[18,20]. A recent phylogeny study suggests that changes in species or subclade of alkenone-producing Isochrysidales appear to affect the relative abundance of alkenone compounds, which may confound the interpretation of alkenone-derived paleoclimatic indicators in saline lakes[29]. However, the evident spatial patterns of $U_{37}^{K'}$ and %$C_{37:4}$ records observed from investigated mid-latitude Asian lakes (Fig. 2, Supplementary Fig. 2) cannot be attributed to species effects due to contrasting associations of $U_{37}^{K'}$ with %$C_{37:4}$ changes in various regions. Thus, we consider that variations in alkenone-based records from

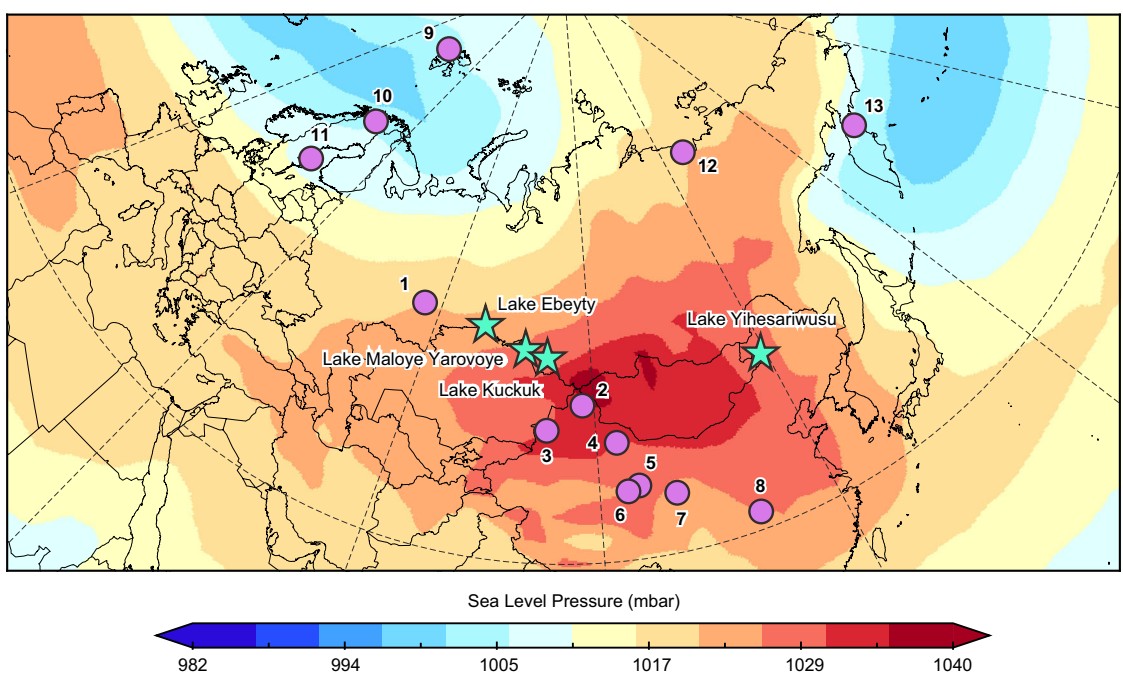

**Fig. 1 | Site location map.** Locations of Lake Yihesariwusu, Ebeyty, Kuchuk, and Maloye Yarovoye (stars, this study) and other temperature records discussed in the main text (dots, site locations listed in Supplementary Data 2). Site 1: Kinderlinskaya Cave, 2: Sahara Sand Peatland, 3: Lake Sayram, 4: Lake Balikun, 5: Lake Hala, 6: Lake Hurleg, 7: Lake Qinghai, 8: Dajiuhu Peatland, 9: Lake Hakluyt and Hajeren, 10: Lake Tsuolbmajavri, 11: Lake Gilltjarnen, 12: Lena River Delta, 13: Lake Pechora. Base map showing winter (DJF) mean sea level pressure for 1991-2020. Climatology data from the NCEP/NCAR Reanalysis at NOAA Physical Sciences Laboratory (http://www.esrl.noaa.gov/psd/).

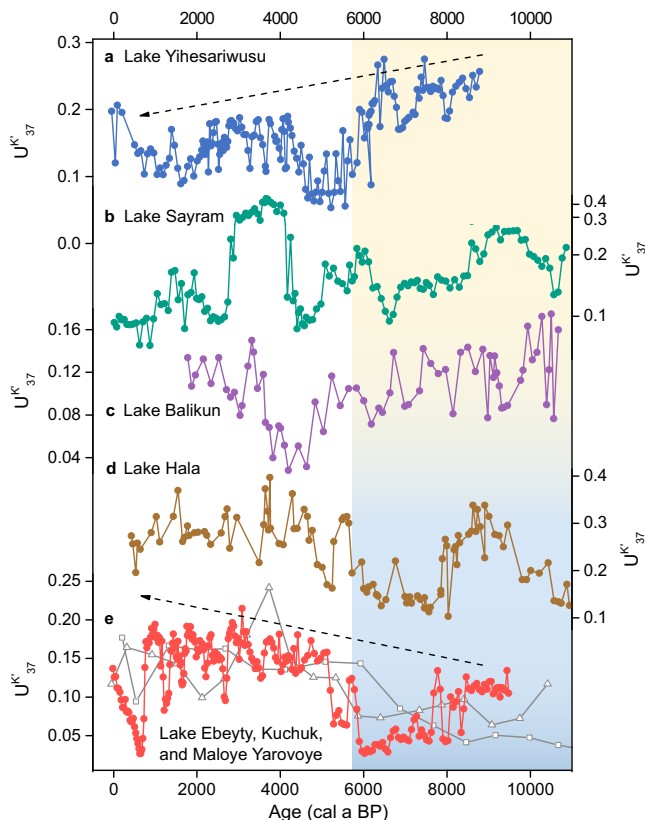

**Fig. 2 | Alkenone unsaturation ($U_{37}^{K'}$) records from mid-latitude Asian lakes.**
**a** Lake Yihesariwusu (this study), **b** Lake Sayram[18], and **c** Lake Balikun[43] in north-western China, **d** Lake Hala on the northeastern Tibetan Plateau[42], and **e** Lake Ebeyty, Kuchuk, and Maloye Yarovoye in southwestern Siberia (dots, squares, and triangles, respectively, this study). See lake locations in Fig. 1. Yellow/blue bar indicates warmer/colder conditions during the early to mid-Holocene.

these lakes primarily indicate regional climatic changes, rather than species factors, during the Holocene.

Lake Yihesariwusu is located in a marginal monsoon region, while Lake Ebeyty, Kuchuk, and Maloye Yarovoye are located in the westerlies-dominated region. Contrasting Holocene moisture changes between the two regions have been widely recognized[30]. Our published $\%C_{37:4}$ records from both regions[18,31,32] are consistent with regional moisture changes inferred from various hydrological indicators[18,32–41]. $\%C_{37:4}$ records from Lake Yihesariwusu and Qinghai[18,31,32] show wettest conditions during the mid-Holocene followed by generally drying trends towards the late Holocene (Supplementary Fig. 3a, b), consistent with hydrological changes inferred from pollen[34,35] (Supplementary Fig. 3c, d), paleosol[36] (Supplementary Fig. 3e), and stabilized dune sites records[37] (Supplementary Fig. 3f) from marginal monsoon regions. $\%C_{37:4}$ records from Lake Ebeyty, Kuchuk, and Maloye Yarovoye in southwestern Siberia and Lake Sayram[18] in northwestern China show overall Holocene wetting trends (Supplementary Fig. 4a, b). Such wetting trends are consistent with hydrological changes inferred from pollen records in the same sediment cores[33,38] (Supplementary Fig. 4c, d) and synthesized moisture records[39,40] (Supplementary Fig. 4e, f) from westerlies-dominated regions. Hence, we suggest that alkenone $\%C_{37:4}$ records from mid-latitude Asian lakes could successfully indicate lake salinity variations and, thus, hydroclimatic changes during the Holocene. The $U_{37}^{K}$ records from mid-latitude Asian lakes, following the structure of $\%C_{37:4}$ variations, could be largely confounded by salinity changes, while lake salinity has little effect on the $U_{37}^{K'}$ index[26–28]. Therefore, we use $U_{37}^{K'}$ index, instead of $U_{37}^{K}$ index, as a warm season temperature indicator in mid-latitude Asian lakes.

## Spatial patterns of temperature changes

$U_{37}^{K'}$ record from Lake Yihesariwusu (Fig. 2a) is characterized by a long-term Holocene cooling trend, with an average value of 0.21 before ~6000 a BP (years before 1950 AD) and 0.14 after that. However, $U_{37}^{K'}$ records from Lake Ebeyty, Kuchuk, and Maloye Yarovoye (Fig. 2e) show an average value of 0.07 before ~6000 a BP and 0.14 after that, indicating extremely low temperature during the early to mid-Holocene and a warmer late Holocene. Such long-term warming trends in southwestern Siberia (Fig. 2e) are in striking contrast to the overall cooling trends from Lake Yihesariwusu (Fig. 2a) and Sayram[18] (Fig. 2b) in northern China, while the warmth at the interval of ~4000–3000 a BP appears to be exceptional at Lake Sayram. Previously reported $U_{37}^{K'}$ records from mid-latitude Asian lakes also exhibit variable long-term trends over the Holocene. Holocene $U_{37}^{K'}$ record from Lake Hala[42] (Fig. 2d), a high-elevation lake on the northeastern Tibetan Plateau, shows a slight warming trend. $U_{37}^{K'}$ record from Lake Hurleg in the Qaidam Basin[18,19] (Supplementary Fig. 5c) does not appear to show discernible long-term trends during the Holocene. $U_{37}^{K'}$ records from Lake Balikun[43] (Fig. 2c) and Lake Qinghai[18,31,32] (Supplementary Fig. 5b) in northern China display a slight cooling trend. Hence, based on the same alkenone proxy, which indicates warm season temperature, a clear Holocene warming trend occurs in southwestern Siberia; such a warming trend disappears approaching northern China and is likely shifted to a cooling trend (Fig. 2).

Holocene temperature records inferred from various proxies also show spatially variable long-term trends over the mid-latitude Eurasian continent (Fig. 3). Branched fatty alcohol[44] (Fig. 3b) and pollen records[45] from central China, chironomid records from Russia Far East[46] (Fig. 3c) and northern Finland[47] (Fig. 3d), and alkenone records from Svalbard[24] (Fig. 3e), representing annual, annual, July, July, and warm season temperatures, respectively, show relatively high annual mean/summer temperatures before ~6000 a BP followed by a general cooling trend. However, an archaeal record from Altai Mountains[48] (Fig. 3g), oxygen isotope records from Siberian Arctic[49] (Fig. 3h) and Ural Mountains[50] (Fig. 3i), and chironomid record from central Sweden[51] (Fig. 3j), representing annual, winter, winter, and summer temperatures, respectively, indicate low seasonal/annual temperatures before ~6000 a BP relative to the late Holocene. Additionally, records used to reconstruct global Holocene temperature changes[2,11,45], inferred from various proxies potentially biased toward different seasons, show that Holocene seasonal/annual cooling trends prevailed in northern China, northern and western Europe (Methods, Fig. 4a), whereas central to eastern Europe and western to central Siberia are broadly characterized by long-term seasonal/annual warming trends (Fig. 4a). Moreover, a synthesis of pollen records reveals Holocene long-term cooling trends in 0–50°N of Asia but warming trends in 50–75°N of Asia in both summer and winter seasons[11] (Fig. 5b, c), and such spatial patterns are broadly consistent with our results (Supplementary Fig. 5b, c). Yet, the climatic transitions from warm to cold (or cold to warm) conditions at the time interval of ~6000–5000 a BP, as indicated by our $U_{37}^{K'}$ records, are not well documented by the synthesized pollen records[11,13] (Fig. 5b, c).

Our alkenone $U_{37}^{K'}$ records, together with published datasets, collectively show that Holocene temperature records inferred from the same proxy, alkenone (Fig. 2, Supplementary Fig. 5), chironomid (Fig. 3c, d, j), or pollen[11] (Fig. 5b, c), display contrasting long-term trends in different regions of the mid-latitude Eurasian continent. Further, temperature records inferred from various proxies that are potentially biased toward different seasons show similar long-term trends in the same regions. For example, $U_{37}^{K'}$ (Fig. 3f) and oxygen isotope records[49,50] (Fig. 3h, i), which are suggested to represent warm and cold season temperatures, respectively, display spatially consistent long-term Holocene warming trends in Siberia. Contrasting Holocene temperature records over the mid-latitude Eurasian continent (Figs. 2, 3, and 4a) are thus unlikely to be attributed to

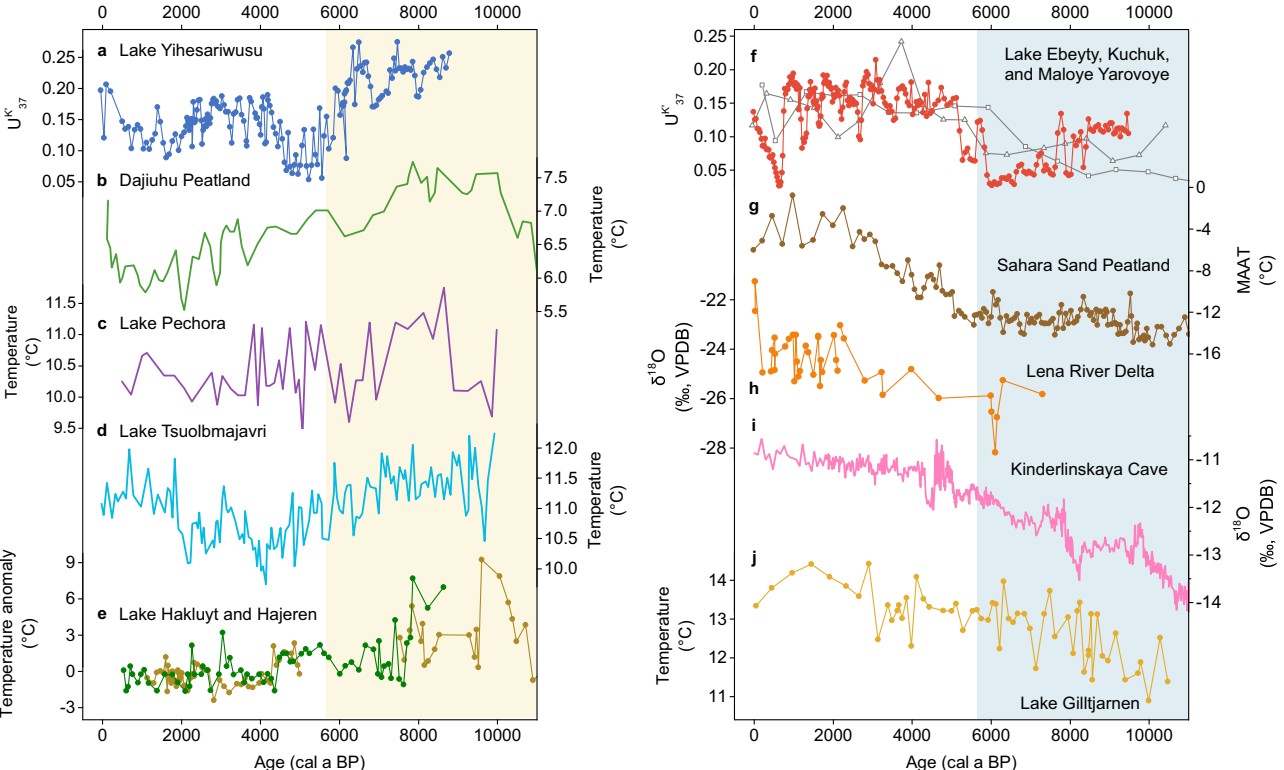

**Fig. 3 | Holocene temperature records over the extratropical Eurasian continent.** Temperature records from (**a**) Lake Yihesariwusu (this study, alkenone unsaturation record, i.e., $U_{37}^{K'}$, warm season temperature), **b** Dajiuhu Peatland (annual mean)[44], **c** Lake Pechora (July)[46], **d** Lake Tsuolbmajavri (July)[47], **e** Lake Hakluyt and Hajeren (warm season)[24], **f** Lake Ebeyty, Kuchuk, and Maloye Yarovoye

(dots, squares, and triangles, respectively, this study, warm season), **g** Sahara Sand Peatland (annual mean)[48], **h** Lena River Delta (winter)[49], **i** Kinderlinskaya Cave (winter)[50], and **j** Lake Gilltjarnen (July)[51]. See site locations in Fig. 1. Yellow/blue bar indicates warmer/colder conditions during the early to mid-Holocene.

seasonality bias in proxies but rather largely reflect spatial patterns of temperature changes. Such spatial patterns appear to persist throughout the year rather than being limited to a particular season (Figs. 3 and 4a). Therefore, we consider that colder airmass appears to have prevailed in the interior of the mid-latitude Eurasian continent during the early to mid-Holocene (Fig. 4a), a spatial feature that has been largely overlooked in previous synthesis studies[1,2].

## Implications for data synthesis and model simulations

A hotly debated issue regarding the Holocene temperature conundrum is the possible seasonality in proxy reconstructions[3–6]. Spatially variable temperature trends inferred from the same proxies in different regions (Figs. 2, 3, and 4a, Supplementary Fig. 5), combined with the replicability of regional temperature trends among various proxy types (Figs. 3 and 4a), provide compelling evidence of the sensibility of proxies to regional temperature variations and reliability of spatial patterns of Holocene temperature changes over the mid-latitude Eurasian continent. Our findings, in conjunction with recent seasonal temperature reconstructions[10,11] (Fig. 5b, c), challenge the hypothesis that the model-data discrepancy might be associated with seasonality bias in proxies[3,4,6,7], at least for the study region. The proposed seasonal origin for the Holocene thermal maximum in global temperature reconstructions, based on the assumption that sea-surface temperature records are primarily driven by local seasonal insolation[4], could not account for the spatial patterns of Holocene terrestrial temperature changes and associated controlling factors. Further, regional temperature synthesis[1,2,11,13], based on latitudinal binning, could also potentially mask the spatial patterns as it tends to smooth out the opposite trends in different regions for the same latitudes. Therefore, a greater focus on geographically distributed temperature reconstructions, along with regional syntheses of

Holocene temperature records, could enable a more comprehensive assessment of dynamical responses to climate forcings, which may provide new insights into the Holocene temperature conundrum.

Several model and data assimilation studies have reported Holocene temperature changes at a regional scale[9,12,52–55]. However, the spatial patterns of Holocene annual temperature changes observed in reanalyzed model data[12], which show marked long-term annual cooling trends over the extratropical Eurasian continent (Fig. 4d), suggested to respond to increased Arctic sea ice coverage, are opposite to what the proxy-based results suggest here (Figs. 2, 3, and 4a). Similarly, recent data assimilations[9,52] fail to capture the substantial temperature discrepancies over the extratropical Eurasian during the mid-Holocene, which show spatially unanimous higher annual mean temperatures over this region at 6000 a BP than 500 a BP (Fig. 4b). Simulated annual mean temperatures from PMIP4-CMIP6 models[53] show lower temperatures over the mid-latitude Asia at 6000 a BP than the preindustrial period, but higher temperatures over the Arctic Ocean (Fig. 4c). Therefore, the spatial patterns over mid-latitude Eurasia identified here are in contrast with those in both model simulations and data assimilation results (Fig. 4). TraCE-21ka simulation results[54] show spatially consistent Holocene summer cooling and annual/winter warming trends in northeastern China and southwestern Siberia (Fig. 5a, d), thus not capturing the spatial patterns observed from proxy-based reconstructions (Fig. 5b, c). The simulated seasonal temperature anomalies generated by multiple climate models[53,55] also indicate spatially unanimous higher summer temperatures and lower winter temperatures over the extratropical Eurasian at 6000 a BP than the preindustrial period, and thus consistent summer cooling and winter warming trends since the mid-Holocene. Current climate models appear to exhibit limitations in simulating the spatial patterns of Holocene temperature changes over the extratropical Eurasian continent.

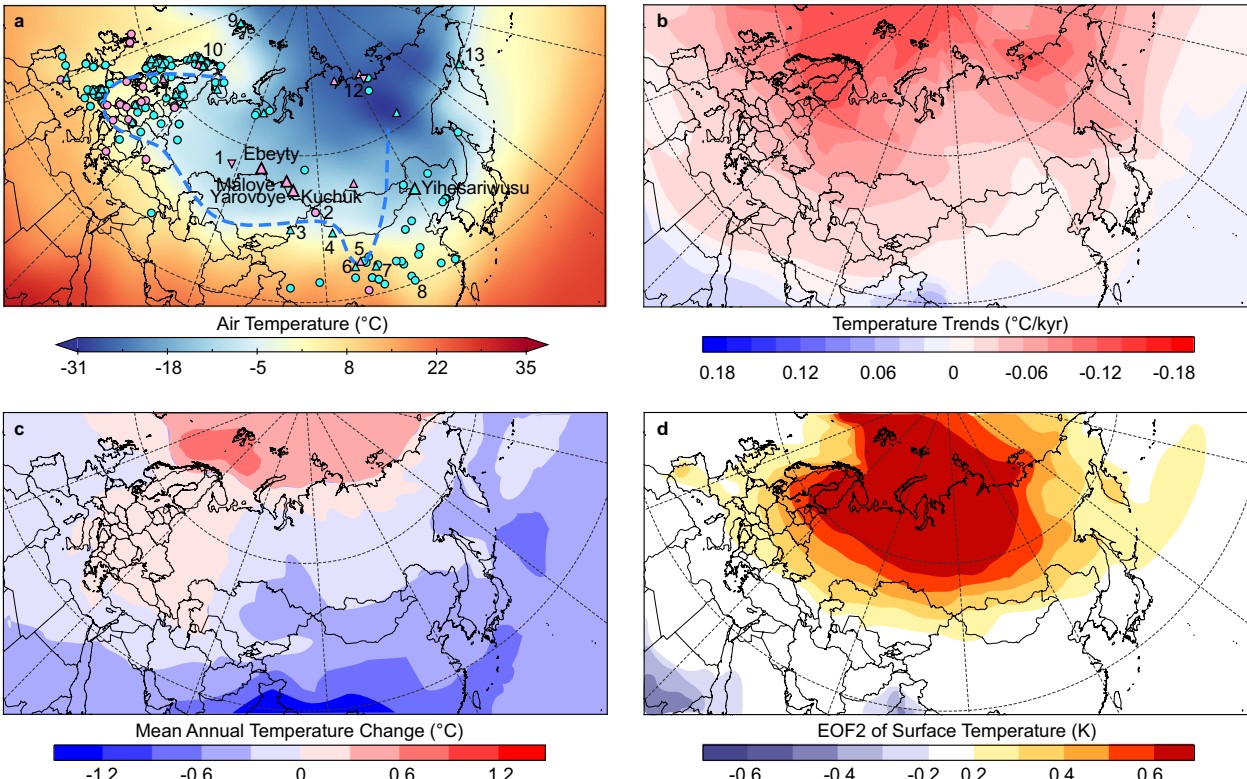

**Fig. 4 | Holocene temperature records over the extratropical Eurasian continent and annual mean temperature changes inferred from data assimilation and model simulations. a** Locations of Holocene terrestrial temperature records from this study and previous studies. Dots, triangles, and inverted triangles represent records indicating annual mean, summer, and winter temperature changes, respectively. Records are listed in Supplementary Data 2. See Methods for classification of records. Numbers refer to site locations in Fig. 1. Pink/blue colors indicate records with long-term warming/cooling trends since ~8000 a BP. The blue line indicates the approximate boundary between the two different temperature patterns. Base map showing winter (DJF) mean air temperature at 1000 hPa for 1991–2020. The source of climatology data is the same as Fig. 1. **b** Annual mean temperature trends from 6000 to 0 a BP (°C kyr⁻¹) inferred from data assimilation[52]. The red/blue color indicates cooling/warming trends and thus higher/lower temperatures during the mid-Holocene than the present. **c** Annual mean temperatures at 6000 a BP compared to 500 a BP (°C) generated by PMIP4-CMIP6 simulation[53]. The red/blue color indicates higher/lower temperatures during the mid-Holocene than 500 a BP. **d** Simulated Holocene cooling mode in a transient simulation[12]. Yellow and red colors indicate regions showing Holocene cooling trends.

Spatial patterns of hydroclimate changes over the mid-latitude Eurasian continent have been well documented throughout the Holocene[30]. Our alkenone %$C_{37:4}$ record from Lake Yihesariwusu displays a relatively dry early Holocene and the wettest mid-Holocene (Methods, Supplementary Fig. 2b), while those from southwestern Siberia show overall Holocene wetting trends (Supplementary Fig. 2e), following the hydrological changes in marginal monsoon regions and westerlies-dominated regions, respectively[30] (Supplementary Figs. 3 and 4). Based on compilations of Holocene climatic records from the Northern Hemisphere, changes in latitudinal temperature gradient are proposed to influence the strength of westerlies flow and, in turn, mid-latitude moisture variability during the Holocene[14]. Reduced mid-latitude annual net precipitation during the early to mid-Holocene is thus attributed to decreased temperature gradient[14]. However, the spatial complexity of temperature and hydrological changes over the mid-latitude Eurasian continent could confound this hypothesis, and hydrological variations in this region, particularly in westerlies-dominated regions, did not correspond to latitudinal temperature gradient changes[14]. Rather, prevailing colder airmass could have controlled regional hydroclimate changes during the early to mid-Holocene.

## Possible controls on spatial patterns

The spatial feature over mid-latitude Eurasia, well identified here (Fig. 4a), remains to be explained physically as it has not been captured by current transient model simulations. Extratropical Northern Hemisphere received enhanced annual insolation during the early to mid-Holocene[56] (Fig. 6a), while the total solar irradiance remained at a relatively low level[57] (Fig. 6b). The spatial patterns of Holocene temperature changes do not appear to be directly associated with decreasing boreal insolation or increasing greenhouse gas concentration (Fig. 6a) because such global-scale forcings could induce spatially consistent temperature variations if other forcings/feedbacks are not involved. Current model simulations[12,53–55] have included changes in both forcings yet are unable to capture the substantial temperature discrepancies over extratropical Eurasia during the early to mid-Holocene (Figs. 4c, d and 6g). Therefore, additional climate forcings or feedbacks are required to explain the spatial patterns of Holocene temperature changes over this region.

Examination of climate forcings or feedbacks inferred from model simulations and climatic records might provide a plausible interpretation of the spatial patterns observed from proxy reconstructions. Supported by model simulations[55], a stable anticyclonic system over northern high latitudes associated with Northern Hemisphere ice sheets could have altered regional atmospheric circulation dynamics and regional climatic conditions during the Last Glacial Maximum[58]. Shown in a coupled atmosphere-ocean model simulation[55], remnant ice sheets during the early to mid-Holocene (Fig. 6c), suggested to contribute to the simulated Holocene annual warming trend[3], could have still exerted an influence on atmospheric circulation dynamics over the mid-latitude Eurasian continent as they could have induced pressure and anticyclonic winds anomalies over northern mid- to high

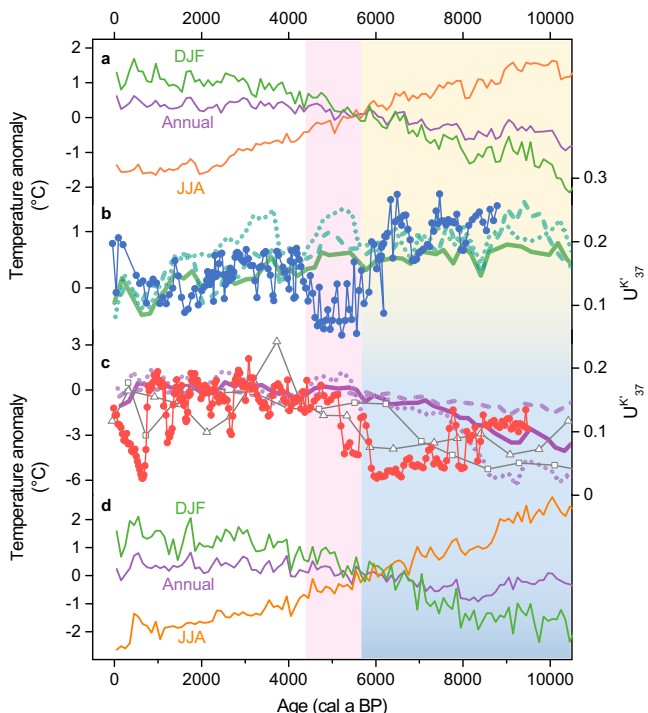

**Fig. 5 | Pollen-based Holocene seasonal temperature records in Asia compared with alkenone unsaturation (U$^{K'}_{37}$) records from this study and CCSM3-simulated temperature anomalies. a** CCSM3-simulated summer (JJA, orange line), winter (DJF, green line), and annual (purple line) temperature anomalies in north-eastern China (46.39°N, 120.00°E, near Lake Yihesariwusu)[54]. **b** Annual (solid green line), summer (dashed green line), and winter (dotted green line) temperature changes in 0–50°N of Asia[11], and U$^{K'}_{37}$ record from Lake Yihesariwusu (dots). **c** Annual (solid purple line), summer (dashed purple line), and winter (dotted purple line) temperature changes in 50–75°N of Asia[11], and U$^{K'}_{37}$ record from Lake Ebeyty, Kuchuk, and Maloye Yarovoye (dots, squares, and triangles, respectively). **d** CCSM3-simulated summer (JJA, orange line), winter (DJF, green line), and annual (purple line) temperature anomalies in southwestern Siberia (53.81°N, 75.00°E, near investigated Siberian lakes)[54]. The yellow/blue bar indicates warmer/colder conditions during the early to mid-Holocene. The pink bar highlights the climatic transition during the mid-Holocene.

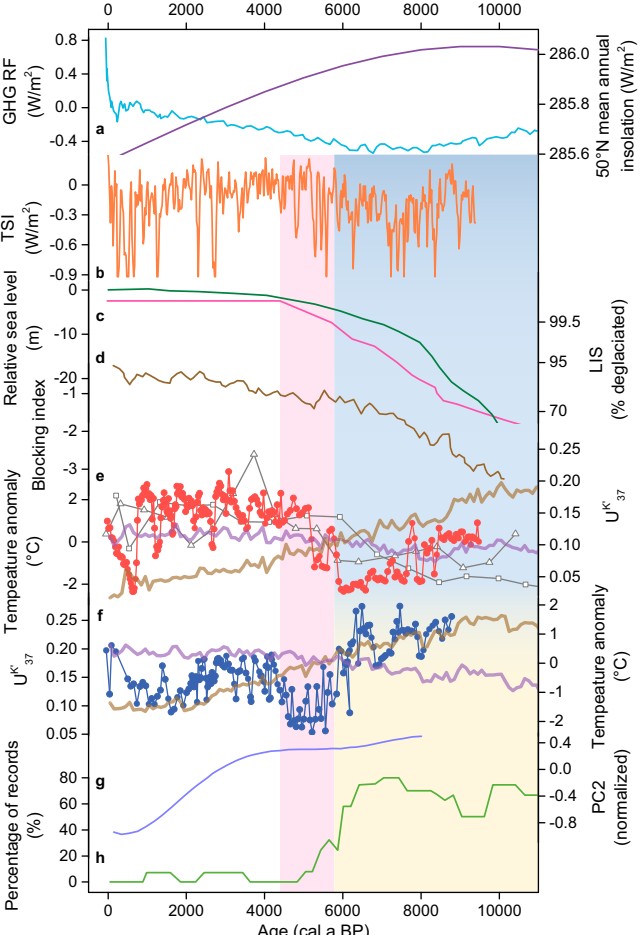

**Fig. 6 | Holocene climate forcings and temperature records. a** Mean annual insolation at 50°N[56] (purple line) and greenhouse gases radiative forcing[70] (GHG RF, blue line). **b** Total solar irradiance (TSI)[57]. **c** Laurentide ice sheet (LIS) deglaciation[71] (pink line) and relative sea-level[72] (green line). **d** Atmospheric blocking index over Scandinavia[50]. **e** Alkenone unsaturation (U$^{K'}_{37}$) records from Lake Ebeyty, Kuchuk, and Maloye Yarovoye (dots, squares, and triangles, respectively), and CCSM3-simulated annual (thick purple line) and summer (JJA, thick brown line) temperature changes in southwestern Siberia (53.81°N, 75.00°E, near investigated Siberian lakes)[54]. **f** U$^{K'}_{37}$ records from Lake Yihesariwusu (dots) and CCSM3-simulated annual (thick purple line) and summer (JJA, thick brown line) temperature changes in northeastern China (46.39°N, 120.00°E, near Lake Yihesariwusu)[54]. **g** Simulated Holocene cooling mode[12]. **h** Sea-ice extent for the Arctic, indicated by the percentage of records showing low sea-ice concentration[60]. The yellow/blue bar indicates warmer/colder conditions during the early to mid-Holocene and associated forcings. The pink bar highlights the climatic transition during the mid-Holocene.

latitudes. A large-scale atmospheric and oceanic reorganization could have occurred in response to the demise of the Laurentide ice sheet and the reduction of meltwater flux[59]. Moreover, a blocking index reconstruction[50] (Fig. 6d) suggests persisted atmospheric blocking over Scandinavia associated with ice sheet forcing during the early to mid-Holocene, causing relaxation of (sub)polar cold airmass and depressed surface temperature in the interior of the mid- to high-latitude Eurasian continent.

Both colder conditions in central to eastern Europe and western to central Siberia inferred from our alkenone results and other temperature records (Figs. 2, 3, and 4a), and the suggested atmospheric circulation dynamics, characterized by the presence of high-pressure system and frequent atmospheric blocking events during the early to mid-Holocene over the mid-latitude Eurasian continent[50,55,59], broadly resemble current winter conditions (Fig. 1). Interestingly, more records showing lower temperatures before ~6000 a BP to the south of Scandinavia than north also resemble current winter conditions (Fig. 4a). Additionally, most Arctic Ocean sea-ice records show less extensive sea–ice cover during the early to mid-Holocene[9,60] (Fig. 6h). Hence, the overall high temperature conditions over the Arctic Ocean, accompanied with low temperature conditions over the central Eurasian continent, during the early to mid-Holocene, might be broadly comparable to the recent Arctic warming and Eurasian cold winters[61,62].

Given the resemblances in atmospheric circulation patterns and temperature conditions over the mid-latitude Eurasian continent during current winters and the early to mid-Holocene, we consider that the enhanced wavy pattern of westerlies, associated with the persistent anticyclone system, perhaps induced by the remnant ice sheets before ~6000 a BP, is a plausible interpretation of spatial patterns of Holocene temperature changes in this region. Cold conditions in the interior of the mid- to high-latitude Eurasian continent, in contrast to the high temperatures in surrounding regions, during the early to mid-Holocene, shaped the spatial patterns of long-term Holocene temperature changes over the mid-latitude Eurasian continent.

As ice sheet-induced anticyclone might have persisted over northern mid-high latitudes[50] (Fig. 6d), we suggest that large-scale changes in land-sea temperature and pressure gradients could induce more meridional, or wavy patterns of westerlies, and thus cold (sub)

polar airmass reaching the interior of continental Eurasia. Consequently, regions affected by cold air masses show depressed surface temperatures during the early to mid-Holocene (Figs. 4a and 6e), and the climatic transition towards a warmer late Holocene occurred in response to the demise of Northern Hemisphere ice sheets (Fig. 6c). Regions outside the influence of (sub)polar air masses should have followed the decreasing boreal insolation, displaying Holocene long-term cooling trends (Fig. 6f). The boundary between the two distinct temperature patterns appears to be roughly situated along northern Xinjiang and may extend to the northeastern Tibetan Plateau (Fig. 4a). Consequently, temperature records from these regions show subtle or indiscernible long-term trends over the Holocene (Fig. 2c, d, Supplementary Fig. 5b, c). The enhanced anticyclone, rather than reduced latitudinal temperature gradient[14], could also well explain drier conditions in the interior of the mid-high latitude Eurasia continent during the early to mid-Holocene, which may further affect hydroclimate changes in marginal monsoon regions (Supplementary Fig. 3).

In summary, here we report spatial patterns of Holocene temperature changes over the mid-latitude Eurasian continent. Colder airmass appears to have prevailed in the interior of the mid-latitude Eurasian continent during the early to mid-Holocene, associated with the wavy pattern of westerlies perhaps modulated by remnant ice sheets, although the exact physical mechanism remains to be explored. Our results challenge the hypothesis of seasonality bias in proxies, at least for this region and the simulated regional spatial patterns of Holocene temperature changes, with important implications for the Holocene temperature conundrum and terrestrial hydroclimate changes.

## Methods
### Materials and chronology
Lake sediment cores were retrieved from Lake Yihesariwusu in northeastern China and Lake Ebeyty, Kuchuk, and Maloye Yarovoye in southwestern Siberia (Fig. 1). Lake Yihesariwusu (48.13°N, 118.63°E) is a hydrologically closed saline lake located in the center of Hulun Buir dune field, China. The lake is mainly recharged by meteoric precipitation, with an area of ~5 km² and a maximum water depth of ~4.0 m[63]. The lake catchment is surrounded by semi-fixed and mobile dunes to the north and sandy land to the south. The mean annual temperature of this region is −1.3 °C and the mean annual precipitation is 349 mm[63]. Two overlapping sediment cores were retrieved from Lake Yiheshariwusu using a piston corer[63]. The original core chronology was established using $^{210}$Pb and thirteen $^{14}$C dates without considering the reservoir ages[63]. Here, we updated the core chronology using an additional $^{14}$C date (Supplementary Data 1) and subducted a reservoir age of 570 years determined by the age difference between plant remains at 14 cm and bulk organic matter at 14.5 cm (Supplementary Data 1). Three anomalously old $^{14}$C ages at 437 cm, 506 cm, and 582 cm may indicate old terrestrial carbon carried by meltwater during the deglaciation to early Holocene[64] and were excluded from the updated depth-age model constructed using Bacon software[65]. The core chronology suggests a basal age of ~8800 a BP (Supplementary Fig. 1a).

Lake Ebeyty (54.65°N, 71.73°E, 50 m, asl) is a hydrologically closed hypersaline lake located in southwestern Siberia, Russia, with an area of 90–113 km² and a periodically changeable depth of 0.6–3 m[66]. The mean annual temperature of this region is 2 °C, and the mean annual precipitation is 420 mm. A ~3.0 m-long core was retrieved from the lake at a water depth of 0.6 m[67]. The core chronology was established using $^{14}$C dates, including 13 reported previously[67] and an additional date (Supplementary Data 1). Two anomalously old $^{14}$C dates below 270 cm depth (Supplementary Data 1) were excluded from the depth-age model constructed using Bacon software[65]. A reservoir age of 700 years was determined by the intercept of linear regression of four $^{14}$C dates from the upper 50 cm of the core. Here, we analyzed samples

above 270 cm, spanning the last ~9600 years based on the depth-age model for the lake (Supplementary Fig. 1b).

Lake Kuchuk (52.69°N, 79.84°E, 98 m, asl) and Maloye Yarovoye (53.03°N, 79.11°E, 96 m, asl) are also located in southwestern Siberia, Russia. Lake Kuchuk is a hydrologically closed hypersaline lake with an area of 166 km² and a maximum depth of 3 m[33]. Lake Maloye Yarovoye, located 40 km north of Lake Kuchuk, is a hydrologically closed hypersaline lake with an area of 35 km² and a maximum depth of 5 m[33]. The mean annual temperature of this region is 0 °C, and the mean annual precipitation is 300 mm[33]. A ~3.1 m-long core was taken from Lake Kuchuk at a water depth of 2.9 m, and a ~5.0 m-long core was taken from the lake center of Maloye Yarovoye at a water depth of 3.6 m[33]. The original core chronologies were established using accelerator mass spectrometry $^{14}$C dates of bulk organic matter[33]. Here, updated reservoir ages of 1000 and 1200 years were determined by the intercepts of linear regressions of $^{14}$C dates from Lake Kuchuk and Maloye Yarovoye, respectively, slightly different from the originally proposed ones[33]. The updated core chronologies suggest a basal age of ~14,000 a BP for the core from Lake Kuchuk and ~12,000 a BP for the core from Lake Maloye Yarovoye. Here, we report records over the Holocene from the two lakes.

### Alkenone analysis
We have analyzed long-chain alkenones in 172 samples from Lake Yihesariwusu, 235 samples from Lake Ebeyty, 19 samples from Lake Kuchuk, and 15 samples from Lake Maloye Yarovoye. Freeze-dried samples were extracted ultrasonically (3 × 30 min) with a solvent of dichloromethane: methanol (9:1, v/v, 15 ml each time). The total lipids were dried under $N_2$ and saponified with 6% KOH in MeOH for at least 12 h at room temperature. Subsequently, 1.5 ml of NaCl in water (5%, w/w) was added, and the neutral solution was extracted with 4 ml n-hexane. The neutral solution was then separated into three fractions with silica gel column chromatography using 4 ml n-hexane, 4 ml dichloromethane, and 4 ml methanol, respectively, while the dichloromethane fraction contains alkenones. The alkenone fraction was analyzed on an Agilent 7890 Gas Chromatography (Agilent DB-1 column: 60 m × 250 μm × 0.10 μm film thickness) with Flame Ionization Detection (FID) at The University of Hong Kong, using n-$C_{36}$ n-alkane as an internal standard for quantification. Each sample was injected in a splitless mode, with $H_2$ as carrier gas. The oven temperature program for alkenone analysis was 60 °C (1 min) to 270 °C at 13 °C /min, and then to 310 °C at 3 °C /min (held 30 min). The laboratory standards were repeatedly analyzed to assess the analytical precision.

The relevant alkenone-based proxies $U^{K'}_{37}$[68] and $\%C_{37:4}$[69] were calculated as follows:

$$U^{K'}_{37} = \frac{[C_{37:2}]}{[C_{37:2}] + [C_{37:3}]} \qquad (1)$$

$$\%C_{37:4} = \frac{[C_{37:4}]}{[C_{37:2}] + [C_{37:3}] + [C_{37:4}]} * 100\% \qquad (2)$$

where $[C_{37:2}]$, $[C_{37:3}]$ and $[C_{37:4}]$ are the concentrations of di-, tri-and tetra-unsaturated $C_{37}$ alkenones, respectively. The estimated analytical error is typically within 0.01 for $U^{K'}_{37}$, and 5% for $\%C_{37:4}$.

### Holocene temperature dataset
Holocene terrestrial temperature datasets over the extratropical Eurasian continent shown in Fig. 4a were obtained from ref. 2,11,45 and listed in Supplementary Data 2. Pollen, GDGTs, and fatty alcohol-based records were (re)calibrated to represent mean annual temperature changes[2,11,45], whereas chironomid and alkenone/oxygen isotope records indicate warm/cold season temperature changes[2,11,45]. Records with locations shown in Fig. 1 are also presented in Fig. 4a. Dots, triangles, and inverted triangles in Fig. 4a represent records indicating

annual mean, summer, and winter temperature changes, respectively. The spatial patterns of Holocene temperature changes over the extratropical Eurasian continent appear to be persistent throughout the year rather than being limited to a particular season. Therefore, we present temperature records inferred from different proxies and compare such patterns with annual mean temperature changes inferred from model and data assimilation in Fig. 4. As a general warming trend has been identified in the early Holocene (~11,000–7000 a BP) over the Eurasian continent[11], we choose to define temperature trends since ~8000 a BP, to focus on spatial patterns of Holocene temperature changes. We classify those records into two types, i.e., warming and cooling types (pink and blue colors in Fig. 4a, respectively) based on their overall trends since ~8000 a BP. Records with insignificant long-term trends are not included in our synthesis. Despite uncertainties in reconstructions and calibrations, it can be observed that records with a warming trend are predominantly concentrated within the interior of the mid-latitude Eurasian continent.

## Data availability

We declare that the data that support the findings of this study are available in the Supplementary Data and can also be accessed at https://doi.org/10.6084/m9.figshare.23731524. All new data associated with the paper are available in the Supplementary Data 1. Lists of the sources of previously published data supporting the findings of this study are available in Supplementary Data 2.

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

## Acknowledgements
This research was supported by the Fund of Shandong Province (LSKJ202203300, to Z.L. and H.W.), Chinese Academy of Sciences (XDB40000000, to W.L. and Z.L.), Hong Kong Research Grants Council Grant (17316322, to Z.L.), State assignment of IGM SB RAS (122041400252-1, to S.K.), National Natural Science Foundation of China (42111530031, to W.L.), and an HKU dissertation year fellowship (to J.J.).

## Author contributions

Z.L., W.L., S.K., and G.C. designed the study. J.J., B.M., and H.W. performed an alkenone analysis. J.J., B.M., H.W., H.L., M.S., Y.H., C.Z., and J.C. performed data investigation. S.K. and G.C. provided samples. Z.L., W.L., and S.K. acquired funding. J.J. and Z.L. led the writing with intellectual contributions from all coauthors.

## Competing interests

The authors declare no competing interests.
