## [Peer Review File NEW · Nature Communications]

Spatial patterns of Holocene temperature changes over mid-latitude EurasiaEditorial Note: Parts of this Peer Review File have been redacted as indicated to remove third-party material where no permission to publish could be obtained.

Reviewer #1 (Remarks to the Author):

This paper presents the Holocene temperature reconstruction based on alkenone UK37 in four lakes in northern Eurasia. The temperatures show opposite trends between southwestern Siberia (three sites in warming trend) and northeastern China (one site in cooling trend). This opposite trend seems to be also consistent with some other temperature proxies of reconstructions. Therefore, the new reconstructions highlight the spatial pattern of the temperature trend in the Holocene. The paper also propose that the contrasting temperature trend is caused by the impact of residual ice on atmospheric circulation. Personally, I think the data analysis itself (including the synthesis with other proxies) is interesting. The argument that the same proxy should represent the same seasonal temperature change is also legitimated (although the comparison with other proxies makes this seasonality messy, because the unknown seasonality of other proxies). The mechanism part is, however, too speculative and confusing, at best. Overall, the paper is poorly presented as it stands. Therefore, I recommend a major revision before it can be published. The major point of the paper, seems to me, is that they proposed a regional pattern of temperature trend in northern Eurasia region and this pattern, if true, remains to be explained physically.

Here are my major concerns.

1. There are many parts of the paper ambiguous about the seasonality. Since the warming/cooling involves potentially seasonality, it is important to be very clear when warming/cooling is stated for both data and model, is it annual mean, or a seasonal change (which season)? In addition, since the Holocene temperature conundrum is for global mean and annual mean temperature, it should also be clear if the temperature is regional or global mean. Here is one example of the ambiguity. (There are too many to list in the paper!)

In the introduction section,

L41: "However, the long-term Holocene warming trend observed in transient climate model simulations is suggested to be forced by retreating Northern Hemisphere ice sheets and rising greenhouse gas concentrations". Is this annual mean, or seasonal, it is global mean or regional?

2. The last section on the mechanism of the regional pattern is, at best, confusing. Indeed, I don't see any physical sensible physical argument, except the speculation of the role of residual ice sheet. For example,

L178: "Remnant ice sheets appear to have exerted a continuous influence on atmospheric circulation dynamics over the mid-latitude Eurasian continent during the early to mid-Holocene as a coupled atmosphere-ocean model simulation shows that they could have induced pressure and anticyclonic winds anomalies over northern mid-high latitudes, and a large-scale atmospheric and oceanic reorganization could have occurred in response to the demise of Laurentide ice sheet and reduction of meltwater flux. Moreover, a blocking index reconstruction (Fig. 4d) suggests persisted atmospheric blocking over Scandinavia associated with ice sheet forcing during the early to mid-Holocene, causing relaxation of polar jet stream and depressed surface temperature in the interior of the mid- to high-latitude Eurasian continent. Both colder conditions in central to eastern Europe and western to central Siberia "

I don't quite understand these arguments. First, do you mean local ice sheet over Eurasia or remote ice sheet from North America? Either way, how much residual ice sheet is left in the Holocene in Eurasia and North America? Second, what is the evidence that the residual ice sheet from North America can cause the contrasting temperature response pattern in northern Eurasia? What caused the change of the blocking index (assuming the reconstruction is correct)? Is it produced by models or your speculation? In either case, it is not obvious why the jet shifts southward in response to the retreating of ice sheet. This southward migration is opposite to the robust results of model experiments on the effect of Laurentide ice sheet at LGM: the retreat of LGM Laurentide ice sheet causes the jet moving northward (from more a southern more zonal position at LGM northward to a northeast-southwest orientation as in the present day), instead of southward. Third, what is the evidence of "a large-scale atmospheric and oceanic reorganization" in response to the residual ice sheet and the associated (small) flux of meltwater? The reorganization in the coupled system is usually discussed for the early deglacial period, when both the ice sheet and the meltwater flux are large.

3. It will be helpful to clarify the point of the paper by presenting some model results. If model shows homogenous warming/cooling for seasonal and/or annual mean, it presents a clear model-

data inconsistency of the regional temperature response. The simplest is to check some regional temperature trend in the available model experiments or data assimilation reanalysis products. Are all of the models generating a uniform temperature change? It will be more fruitful if the authors collaborate with some modelers for the model-data comparison. The model result of Extended Data Fig.5b seems to me not much relevant to the point of the paper. This model is used, not because it explains the physical mechanism, but because it looks like what the authors want.

Minor points:

"A spatiotemporal analysis of annual temperature variability using data from model simulations suggests a Holocene cooling mode over the Arctic Ocean and mid-to-high latitude Eurasia . A data assimilation reconstruction combining proxy records and simulation results also yields warmer conditions across continental Eurasia during the mid-Holocene relative to the preindustrial era" Be consistent, or at least very careful, in specifying the trend of cooling/warming against the colder/warmer condition in the mid-Holocene. They are of the opposite signs. For example, the sentence above is unambiguous. The first part "cooling" mode seems to refer to the cooling towards late Holocene (so it is warmer condition in the mid-Holocene). The second part "warmer condition during the mid-Holocene" also implies a cooling towards late Holocene. Are they the same or not? It may be more clear to use either the trend or the mid-Holocene condition relative to the present throughout the paper.

"Yet, a recent synthesis of pollen records indicates a warm early to mid-Holocene followed by generally cooling trends in southern Asia and Europe, but long-term Holocene warming trends in northern high-latitude Asia. Given the uncertainties in regional temperature changes and poorly constrained climate responses to various forcings, the Holocene temperature conundrum remains highly contentious, and our understanding of Holocene hydroclimatic changes and associated mechanisms remains largely speculative"

It appears the authors try to say models show the same sign of cooling towards late Holocene, but a recent synthesis shows opposite trend in different regions in Eurasia continent. That is fine, but i) this point should be clarified explicitly, ii) this is not much relevant to the conundrum, which is about global mean and annual mean.

Extended Data Fig.4: What does it mean by "southern Asia" here? Lake Yihesariwusu is in northeastern China, which is NOT the southern Asia we usually refer to (or I never heard this region is called southern Asia). Be careful on your regional reference. In the region of your interest "northern Eurasia", this is the "southern region", while southern Siberia is the "northern region". You can predefine these regions first (give approximate latitude/longitude ranges...).

L99- "warm conditions prevailed in northern China, northern and western Europe (Fig. 3a)". This description is confusing. To my eye, the warming is dominant in central Europe, while cooling occurs in the surrounding area. Certainly, northern and western Europe are dominated by cooling (blue), instead of warming. It seems the authors claimed this pattern in Europe because they want to show a coherent pattern. But, the European pattern does not seem to me related to the pattern of their lake sites. Again, are these temperature change of seasonal (which season if yes) or annual mean?

L142 "However, the spatial patterns of Holocene temperature changes observed in reanalyzed model data , which show marked long-term cooling trends over the extratropical Eurasian continent (Fig. 4g, Extended Data Fig. 5b), suggested to respond to increased Arctic sea ice coverage, are opposite to what the proxy-based results suggest (Fig. 2, 3)." Again, is Fig.3g for annual mean or seasonal? If your data is seasonal, why don't you compare the seasonal temperature? Same when other models/data assimilation products are mentioned. Refer to my major comment 3 on this.

L170 "The spatial patterns of Holocene temperature changes do not appear to be directly associated with decreasing boreal insolation or increasing greenhouse gas concentration (Fig. 4a), because such global-scale forcings could induce spatially consistent temperature variations." It is not obvious these two mechanisms should be discarded. Is it possible different mechanism is dominated in different regions?

L196: "...might be broadly comparable to recent Arctic amplification and Eurasian cold winters". This does not seem to me a good analogy. Recent arctic amplification is thought to be caused by the rising CO₂, and the warming tends to occur uniformly. The Eurasian cooling is also for winter, opposite to the warm season proxy here. How are these consistent?

203: "As ice sheet-induced anticyclone might have persisted over northern mid-high latitudes (Fig. 4d), we suggest that large-scale changes in land-sea temperature and pressure gradients could induce more meridional, or wavy patterns of westerlies, and southward deflection of the polar front to continental Eurasia". What cause this southward shift, orbital, which season? CO₂? Ice sheet?

Should Ref 7 be Erb et al., 2022?

This ref. 7 (Osama) is for Last Glacial Maximum Reanalysis, and shows a warming trend in the Holocene. Erb et al. is on the Holocene with multiple models and does not show warming trend.

Erb, M. P. et al. Reconstructing Holocene temperatures in time and space using paleoclimate data assimilation. *Clim. Past* 18, 2599–2629 (2022).

Fig.2 and 3, each curve, include the lake name for each label, e.g. a) Lake.....b)Lake.....This helps readers...not have to go back to the caption.

There is no Supplementary Table 2, referred in the caption of Fig.3. There is only supplementary data set 2. But that is not what a reviewer would like to see. A reviewer just needs the type, location et al of each record used.

Reviewer #2 (Remarks to the Author):

In this manuscript, the authors present alkenone based temperature reconstructions for the Holocene: one from the monsoonal marginal zone and three from the continental/westerlies realm. By contextualization of the data they shape out two different spatial patterns of temperature changes throughout the study area and covered time period.

The overall aim of the manuscript is to solve question connected to the „Holocene conundrum“. A recent study (Herzschuh et al 2023; <https://cp.copernicus.org/articles/19/1481/2023/>) highlighted the importance of focussing on spatial Holocene temperature patterns and their regional drivers, for the understanding of this proxy-model mismatch. By further investigating the situation in Asia, this manuscript by Jiang et al, is an important and interesting contribution to the debate. Despite these aspects, I´m not convinced that this study is suitable for NCOMM for the following reasons:

1. I miss a more critical discussion of the alkenone proxy which is often problematic in lakes. I´m not familiar with all of the cited original studies including those of which data from other lakes were plotted, but it appears that many factors can influence the different kinds of uk37-proxies. Salinity changes, translocation of alkenone producers, glacial meltwater influence, community changes of alkenone producers,... are just some of the factors which are mentioned in some of the cited papers. Partially, this is solved in the manuscript by comparison with independent proxies, but I still miss a discussion on how reliable alkenone-based temperature proxies are in the specific case, not only in the methods but in the main part of the manuscript.
2. Connected with these aspects, I find the recent CP-paper by Herzschuh et al is more convincing, because solely focussing on pollen transfer functions, which have been proven to work well in context of precipitation and temperature reconstructions. Unfortunately, this also takes away a bit of novelty from this new manuscript by Jiang et al, especially since Herzschuh et al already showed some spatial heterogeneity of Holocene temperature in Asia.

Concluding, I think this new manuscript is definitely be worth to be published after revisions, but more likely in a journal such as QSR or CP or maybe COMM EARTH ENV.

Minor comments:

- The above mentioned paper by Herzsuh needs to be cited
- Maps in Fig 3a and Ext Fig 5a would benefit from labelling the dots (as in Fig. 1)
- There is a lot of reference to figures in the appendix. I would recommend to reconsider if some of those figures could be shifted to the main manuscript, in case this manuscript is re-arranged to a less short-format version.

We have carefully followed the editor's and reviewers' comments revising our manuscript. In order to clearly highlight the changes made in the revision, we have copied all these comments below in black and inserted our replies in blue. Line numbers mentioned in our response letter refer to the ones in our revised manuscript.

Reviewer #1 (Remarks to the Author):

This paper presents the Holocene temperature reconstruction based on alkenone UK37 in four lakes in northern Eurasia. The temperatures show opposite trends between southwestern Siberia (three sites in warming trend) and northeastern China (one site in cooling trend). This opposite trend seems to be also consistent with some other temperature proxies of reconstructions. Therefore, the new reconstructions highlight the spatial pattern of the temperature trend in the Holocene. The paper also propose that the contrasting temperature trend is caused by the impact of residual ice on atmospheric circulation. Personally, I think the data analysis itself (including the synthesis with other proxies) is interesting. The argument that the same proxy should represent the same seasonal temperature change is also legitimated (although the comparison with other proxies makes this seasonality messy, because the unknown seasonality of other proxies). The mechanism part is, however, too speculative and confusing, at best. Overall, the paper is poorly presented as it stands. Therefore, I recommend a major revision before it can be published. The major point of the paper, seems to me, is that they proposed a regional pattern of temperature trend in northern Eurasia region and this pattern, if true, remains to be explained physically.

We appreciate Reviewer's comments and have revised our manuscript accordingly. We understand Reviewer's concerns regarding the seasonality of proxies and the comparison of different proxies, which are hotly debated in paleoclimatic studies. Temperature records from the interior of the mid-latitude Eurasian continent inferred from different proxies suggested to reflect seasonal/annual mean temperature changes, e.g., oxygen isotope records from Kinderlinskaya Cave and Lena River Delta (revised Fig. 3h, 3i, winter temperature), GDGTs record from Sahara Sand Peatland (revised Fig. 3g, annual mean temperature), and several pollen records from this region (revised Fig. 4a, annual mean temperature), display similar Holocene warming trends as our alkenone-based warm season temperature records from Siberian lakes (revised Fig. 3g). Our results, together with published temperature records, suggest that the spatial patterns of Holocene temperature changes over the mid-latitude Eurasian continent appear to persist throughout the year, rather than being limited to a particular season.

Moreover, ~75% of the temperature records presented in revised Fig. 4a are inferred from (re)calibrated pollen reconstructions, which do not show substantial differences in seasonal and annual mean Holocene temperature trends. Initially, we plotted summer (inferred from pollen, alkenone, and chironomid records) and annual mean (inferred from pollen and GDGTs records) temperature records separately, but the spatial patterns were broadly similar. This again indicates that the spatial patterns inferred from proxy-based records are not limited to a particular season. Therefore, we presented annual mean temperature changes inferred from

pollen and several GDGTs records, and seasonal temperature changes inferred from other proxies, in revised Fig. 4a.

We have included seasonality information of the records discussed in this study in Supplementary Table 2. We have made the following modifications to better address Reviewer's concern about seasonality:

- (1) We have added seasonality information of records to the caption of revised Fig. 3 and revised main text (Line 148-150, 152-154).
- (2) We have modified previous Fig. 3a (revised Fig. 4a) to display records representing different seasons with different symbols.
- (3) We have described the reasons for presenting temperature records inferred from different proxies in revised Fig. 4a (Line 383-388).
- (4) We have compared seasonal/annual temperature changes inferred from models with proxy-based records/spatial patterns in revised main text (Line 206-220) and revised Fig. 4-6.

Modification of the mechanism section is referred to our response to Major Comment #3.

Here are my major concerns.

1. There are many parts of the paper ambiguous about the seasonality. Since the -warming/cooling involves potentially seasonality, it is important to be very clear when warming/cooling is stated for both data and model, is it annual mean, or a seasonal change (which season)? In addition, since the Holocene temperature conundrum is for global mean and annual mean temperature, it should also be clear if the temperature is regional or global mean. Here is one example of the ambiguity. (There are too many to list in the paper!)

In the introduction section,

L41: "However, the long-term Holocene warming trend observed in transient climate model simulations is suggested to be forced by retreating Northern Hemisphere ice sheets and rising greenhouse gas concentrations". Is this annual mean, or seasonal, is it global mean or regional?

The long-term Holocene warming trend observed in transient climate model simulations is for global annual temperature.

We appreciate Reviewer's comments and have carefully clarified our description about seasonality of records and regional/global mean of temperature in Abstract (Line 23, 27), Introduction (Line 44, 59, 60, 62, 64, 66), and Results and discussion (Line 141, 148, 149, 152, 153, 156, 158, 202, 203, 208, 209, 214, 216).

2. The last section on the mechanism of the regional pattern is, at best, confusing. Indeed, I don't see any physical sensible physical argument, except the speculation of the role of residual ice sheet. For example,

L178: “Remnant ice sheets appear to have exerted a continuous influence on atmospheric circulation dynamics over the mid-latitude Eurasian continent during the early to mid-Holocene as a coupled atmosphere-ocean model simulation shows that they could have induced pressure and anticyclonic winds anomalies over northern mid-high latitudes, and a large-scale atmospheric and oceanic reorganization could have occurred in response to the demise of Laurentide ice sheet and reduction of meltwater flux. Moreover, a blocking index reconstruction (Fig. 4d) suggests persisted atmospheric blocking over Scandinavia associated with ice sheet forcing during the early to mid-Holocene, causing relaxation of polar jet stream and depressed surface temperature in the interior of the mid- to high-latitude Eurasian continent. Both colder conditions in central to eastern Europe and western to central Siberia “

I don't quite understand these arguments. First, do you mean local ice sheet over Eurasia or remote ice sheet from North America? Either way, how much residual ice sheet is left in the Holocene in Eurasia and North America? Second, what is the evidence that the residual ice sheet from North America can cause the contrasting temperature response pattern in northern Eurasia? What caused the change of the blocking index (assuming the reconstruction is correct)? Is it produced by models or your speculation? In either case, it is not obvious why the jet shifts southward in response to the retreating of ice sheet. This southward migration is opposite to the robust results of model experiments on the effect of Laurentide ice sheet at LGM: the retreat of LGM Laurentide ice sheet causes the jet moving northward (from more a southern more zonal position at LGM northward to a northeast-southwest orientation as in the present day), instead of southward. Third, what is the evidence of “a large-scale atmospheric and oceanic reorganization” in response to the residual ice sheet and the associated (small) flux of meltwater? The reorganization in the coupled system is usually discussed for the early deglacial period, when both the ice sheet and the meltwater flux are large.

The spatial patterns of Holocene temperature changes over mid-latitude Eurasia, as inferred from our alkenone results and published temperature records, are not captured by the transient model simulations (e.g., Liu et al., 2014; Bader et al., 2020), although model results (Liu et al., 2014) do suggest that retreating ice sheets contributed to the global warming trend over the Holocene. Furthermore, we do not find evidence to suggest that orbital forcing and atmospheric greenhouse gases play different roles in temperature changes in subregions of the extratropical Eurasian continent. Therefore, it seems unlikely that changes in global-scale forcings included in current models can explain the spatial patterns of Holocene temperature changes. Additional climate forcings or feedbacks are required to explain the spatial patterns of Holocene temperature changes over mid-latitude Eurasia. To some degree, we agree with Reviewer's assessment that the spatial pattern, which we believe is clearly identified in our present study, remains to be explained physically. However, we still wish to offer some speculations.

Examination of climate forcings or feedbacks inferred from model simulations and climatic records could provide a plausible interpretation of the spatial patterns observed from proxy reconstructions that have not been captured by current transient models. Model studies (Harrison et al., 1992; Siebert and Dowdeswell, 2004) have reported the presence of anticyclonic circulation over the Fennoscandian sector of the Scandinavian ice sheet during expanded ice extent, which depressed surface temperatures over western Eurasia before the Holocene. Results from the GENMOM coupled atmosphere-ocean model suggest that an ice-

sheet-induced perturbation of atmospheric circulation dynamics was still present during the early to mid-Holocene, producing positive/negative 500-hPa geopotential height and sea-level pressure anomalies at northern mid-high latitudes, despite the reduced Eurasia ice sheets (Alder and Hostetler, 2015). Therefore, we suggest that the atmospheric circulation dynamics over the mid-latitude Eurasian continent appears to be still influenced by remnant Eurasia ice sheet over Scandinavian during the early to mid-Holocene, although, based on current knowledge, the amount of residual ice sheet at that period remains unclear. We insert here (below) the figure by Alder and Hostetler (2015) to show the GENMOM-simulated seasonal 500 hPa geopotential height and wind anomalies relative to PI.

Meanwhile, the blocking index by Baker et al. (2017) is reconstructed based on the $\delta^{18}\text{O}$ gradient between DYE-3 and Renland ice cores, because atmospheric blocking in the North Atlantic sector and Scandinavia produces positive $\delta^{18}\text{O}_{\text{precipitation}}$ anomalies in the eastern relative to southern Greenland. The blocking index correlates strongly with the $\delta^{18}\text{O}$ record from Kinderlinskaya Cave, which is interpreted as a winter temperature indicator. The larger $\delta^{18}\text{O}$ gradient during the early Holocene is explained by the persistence of blocking near Scandinavia that formed in response to the ice-sheet forcing, and therefore depressed surface temperatures over western Eurasia (Baker et al., 2017).

Based on evidence presented above, and resemblances in atmospheric circulation dynamics and temperature conditions over the mid-latitude Eurasian continent during current winters and the early to mid-Holocene, we suggest that enhanced wavy pattern of westerlies associated with the presence of high-pressure system and frequent atmospheric blocking events over the mid-latitude Eurasian continent before ~6,000 a BP appears to be the most plausible interpretation for the proposed spatial patterns of Holocene temperature changes. The wavy pattern of westerlies suggested in our study is not equivalent to the southward migration of jet, but rather a more meridional pattern of westerlies that brings cold (sub)polar air mass reaching the interior of the mid-high latitude Eurasian continent, and thus depressed surface temperatures over this region during the early to mid-Holocene. However, regions outside the influence of cold air masses should have followed the decreasing boreal insolation, displaying Holocene long-term cooling trends. We insert here (below) the figure by Siegert and Dowdeswell (2004) to show the wavy pattern of westerlies generated by an Atmospheric General Circulation Model.

Redacted

Wassenburg et al. (2016) compared Holocene rainfall record from northwest Africa based on speleothem $\delta^{18}\text{O}$ with a speleothem-based rainfall record from Europe. They suggested that the two records are positively correlated during the early Holocene, followed by a shift to an anti-correlation during the mid-Holocene which is similar to the modern condition. Based on simulation results generated by an Earth system model, they proposed that the shift to the anti-correlation reflects a large-scale atmospheric and oceanic reorganization in response to the demise of the Laurentide ice sheet and a strong reduction of meltwater flux. That study showed that atmospheric circulation dynamics during the early to mid-Holocene could have been different from the late Holocene. Stager and Mayewski (1997) also suggested that atmospheric circulation dynamics changed abruptly during the early to mid-Holocene, based on abrupt changes in paleoclimatic records from equatorial East Africa, Antarctica, and Greenland which indicates a large-scale climatic reorganization to full postglacial conditions. Based on results from simulations and reconstructions, we consider that residual ice sheet and meltwater, together with changes in orbital forcing and land-sea thermal differences, could have influenced the atmospheric circulation dynamics during the early to mid-Holocene, and the atmospheric and climatic reorganization occurred after the demise of ice sheet and reduced meltwater.

We understand Reviewer's concern about the physical argument of our suggested/speculated mechanism. The spatial patterns of Holocene temperature changes over mid-latitude Eurasia have not been captured by current transient model simulations, thereby limiting our ability to verify the proposed mechanism through modeling. By integrating anomalies of climate forcings or feedbacks inferred from model simulations and climatic records, and comparing circulation and temperature conditions during the early to mid-Holocene with modern winter conditions, we could offer a plausible interpretation for the spatial patterns of Holocene temperature changes. We agree with Reviewer that the proposed mechanism needs to be physically validated in future studies. At present, our objective is to identify the spatial patterns

of Holocene temperature changes over mid-latitude Eurasia, argue against the seasonality effects on such patterns (at least in the study region), and present a plausible interpretation for further studies.

We have revised the relevant section to incorporate Reviewer's comments. We have clarified that the spatial patterns of Holocene temperature changes over mid-latitude Eurasia, not captured by current simulations, need to be explained with additional forcings/feedbacks (Line 239-240, 242-248). We have softened our wording about the suggested mechanism (Line 31, 34, 251-253, 280-281, 305-306). We have modified our wording regarding cold airmasses reaching the interior of the mid-high latitude Eurasian continent (Line 263-264, 287-288).

3. It will be helpful to clarify the point of the paper by presenting some model results. If model shows homogenous warming/cooling for seasonal and/or annual mean, it presents a clear model-data inconsistency of the regional temperature response. The simplest is to check some regional temperature trend in the available model experiments or data assimilation reanalysis products. Are all of the models generating a uniform temperature change? It will be more fruitful if the authors collaborate with some modelers for the model-data comparison. The model result of Extended Data Fig.5b seems to me not much relevant to the point of the paper. This model is used, not because it explains the physical mechanism, but because it looks like what the authors want.

We agree with Reviewer that model-data comparison helps to clarify our statement. This is also the reason we presented our previous Supplementary Fig. 5, which we meant to show that the simulated pattern (Bader et al., 2020) is OPPOSITE to the one identified in our present study. Meanwhile, we have invited Dr Jun Cheng, who is now our coauthor, to help model-data comparison.

As suggested, we have compared our alkenone-based warm season temperature records with CCSM3-simulated Holocene JJA temperature changes (Liu et al., 2009) in northeastern China and southwestern Siberia (revised Fig. 6e, f). Model results show summer cooling trends in both regions, contrasting with our alkenone records. We have also compared CCSM3-simulated JJA, DJF, and annual temperature anomalies (Liu et al., 2009, revised Fig. 5a, d) with pollen-based seasonal temperature records (Zhang et al., 2022, revised Fig. 5b, c). Model results again indicate similar seasonal/annual mean temperature trends in northeastern China and southwestern Siberia, i.e., long-term winter/annual warming and summer cooling trends. The spatial pattern inferred from pollen synthesis, i.e., long-term seasonal/annual cooling trends in 0–50°N of Asia but warming trends in 50–75°N of Asia, have not been captured by model simulations.

Seasonal surface air temperature anomalies generated by GENMOM, PMIP3, and PMIP4 models all exhibit consistent spatial patterns, indicating higher summer temperatures but lower winter temperatures over the extratropical Eurasian continent at 6,000 a BP compared to the pre-industrial era (Alder and Hostetler, 2015; Brierley et al., 2020). Recent data assimilations (Erb et al., 2022; Kaufman et al., 2023, revised Fig. 4b) show spatially unanimous higher annual mean temperatures over the extratropical Eurasian continent at 6,000 a BP than 500 a

BP, failing to capture the substantial temperature discrepancies over this region during the mid-Holocene. Annual mean surface air temperature changes generated by the PMIP4 model indicate lower temperatures over the mid-latitude Asia at 6,000 a BP compared to the pre-industrial era (Brierley et al., 2020, revised Fig. 4c). Such model-based spatial pattern is in contrast to results derived from data assimilations and proxy reconstructions. We recognize that conducting model–data comparisons could help us gain a better understanding of spatial patterns of Holocene temperature changes and associated mechanisms. We also plan to collaborate with modelers to conduct model-based climate sensitivity experiments to test the speculated mechanisms, but it is expected to be a long-term effort.

We have collaborated with a paleomodeler and expanded the model-data comparison section in the revised main text to better address Reviewer’s concerns. We have incorporated CCSM3-simulated regional temperature changes into our discussion (Line 213-216) and compared proxy-based records with CCSM3-simulated results in revised Fig. 5, 6. Additionally, we have incorporated more data assimilation and model results into our discussion (Line 209-216, 216-220). We have included annual mean surface air temperature conditions at 6,000 a BP compared to the pre-industrial period generated by data assimilation (Erb et al., 2022) and PMIP4 model (Brierley et al., 2020) in revised Fig. 4, which highlights the contrasting spatial patterns between proxy-based reconstructions and data assimilation/model results.

Minor points:

“A spatiotemporal analysis of annual temperature variability using data from model simulations suggests a Holocene cooling mode over the Arctic Ocean and mid-to-high latitude Eurasia. A data assimilation reconstruction combining proxy records and simulation results also yields warmer conditions across continental Eurasia during the mid-Holocene relative to the preindustrial era”

Be consistent, or at least very careful, in specifying the trend of cooling/warming against the colder/warmer condition in the mid-Holocene. They are of the opposite signs. For example, the sentence above is unambiguous. The first part “cooling” mode seems to refer to the cooling towards late Holocene (so it is warmer condition in the mid-Holocene). The second part “warmer condition during the mid-Holocene” also implies a cooling towards late Holocene. Are they the same or not? It may be more clear to use either the trend or the mid-Holocene condition relative to the present throughout the paper.

The “cooling mode” indicates long-term Holocene cooling trends. The “warmer condition during the mid-Holocene”, inferred from the comparison of temperatures at 6,000 a BP with the preindustrial era, indicates cooling trends since the mid-Holocene. We have modified our description of temperature trends here (Line 55-60) and throughout the paper.

“Yet, a recent synthesis of pollen records indicates a warm early to mid-Holocene followed by generally cooling trends in southern Asia and Europe, but long-term Holocene warming trends in northern high-latitude Asia. Given the uncertainties in regional temperature changes and

poorly constrained climate responses to various forcings, the Holocene temperature conundrum remains highly contentious, and our understanding of Holocene hydroclimatic changes and associated mechanisms remains largely speculative”

It appears the authors try to say models show the same sign of cooling towards late Holocene, but a recent synthesis shows opposite trend in different regions in Eurasia continent. That is fine, but i) this point should be clarified explicitly, ii) this is not much relevant to the conundrum, which is about global mean and annual mean.

Thanks for this suggestion and we have modified this part accordingly (Line 60-70).

Extended Data Fig.4: What does it mean by “southern Asia” here? Lake Yihesariwusu is in northeastern China, which is NOT the southern Asia we usually refer to (or I never heard this region is called southern Asia). Be careful on your regional reference. In the region of your interest “northern Eurasia”, this is the “southern region”, while southern Siberia is the “northern region”. You can predefine these regions first (give approximate latitude/longitude ranges...).

Zhang et al. (2022) used southern/northern Asia to refer 0–50°N/50–75°N of Asia. We have provided latitude ranges of these regions to make our description clearer (Line 60-63, caption of revised Fig. 5).

L99- “warm conditions prevailed in northern China, northern and western Europe (Fig. 3a)”. This description is confusing. To my eye, the warming is dominant in central Europe, while cooling occurs in the surrounding area. Certainly, northern and western Europe are dominated by cooling (blue), instead of warming. It seems the authors claimed this pattern in Europe because they want to show a coherent pattern. But, the European pattern does not seem to me related to the pattern of their lake sites. Again, are these temperature change of seasonal (which season if yes) or annual mean?

The sentence Reviewer mentioned is “warm conditions prevailed in northern China, northern and western Europe during the early to mid-Holocene”, which refers to records showing relatively high temperatures during the early to mid-Holocene and long-term Holocene cooling trends, i.e., blue dots in previous Fig. 3a (revised Fig. 4a). It seems that our understanding of the cooling trends in northern and western Europe aligns with what Reviewer has observed. We have modified our wording regarding “warming/cooling conditions during the early to mid-Holocene” and “Holocene warming/cooling trends” here (Line 155-159) and throughout the paper.

We also note that several records from eastern Europe show Holocene cooling trends (blue dots), not consistent with our proposed spatial pattern. However, records indicating Holocene warming trends concentrate within the interior of the mid-latitude Eurasian continent, and more records showing lower temperatures before ~6,000 a BP to the south of Scandinavia than north. Given the uncertainties in proxy reconstructions and temperature (re)calibrations, our assessment regarding cold conditions over the interior of the mid-latitude Eurasian continent

during the early to mid-Holocene appears to be reasonable. Meanwhile, we have softened our wording in the revised main text (Line 158-159).

We have presented records representing different seasons with different symbols in revised Fig. 4a.

L142 “However, the spatial patterns of Holocene temperature changes observed in reanalyzed model data, which show marked long-term cooling trends over the extratropical Eurasian continent (Fig. 4g, Extended Data Fig. 5b), suggested to respond to increased Arctic sea ice coverage, are opposite to what the proxy-based results suggest (Fig. 2, 3).” Again, is Fig. 3g for annual mean or seasonal? If your data is seasonal, why don’t you compare the seasonal temperature? Same when other models/data assimilation products are mentioned. Refer to my major comment 3 on this.

Our previous Fig. 3g (revised Fig. 3f) shows alkenone-based warm season temperature records from Siberian lakes. However, the spatial patterns of Holocene temperature changes over mid-latitude Asia appear to persist throughout the year, rather than being limited to the warm season, as mentioned at the beginning of this letter. Therefore, we compared proxy-based records inferred from different proxies with model-simulated annual mean temperature changes in revised Fig. 4. Meanwhile, we also compared simulated seasonal temperature records with proxy-based records in revised Fig. 5, 6.

L170 “The spatial patterns of Holocene temperature changes do not appear to be directly associated with decreasing boreal insolation or increasing greenhouse gas concentration (Fig. 4a), because such global-scale forcings could induce spatially consistent temperature variations.”

It is not obvious these two mechanisms should be discarded. Is it possible different mechanism is dominated in different regions?

The spatial patterns of Holocene temperature changes over the extratropical Eurasian continent suggested in our study, not captured by model simulations or data assimilation studies, appear to be difficult to explain with changes in boreal insolation or greenhouse gas concentration, as both factors have been included in current models. Currently, it is difficult to find evidence that insolation and greenhouse gas concentration play different roles in temperature changes in subregions of extratropical Eurasia.

Based on the ice-sheet influence on atmospheric circulation dynamics suggested in model simulation and reconstruction studies, in addition to the resemblances in atmospheric circulation dynamics and temperature conditions over the mid-latitude Eurasian continent during current winters and the early to mid-Holocene, we consider that the enhanced wavy pattern of westerlies, associated with the persistent anticyclone system perhaps induced by the remnant ice sheets before ~6,000 a BP, appear to be the most plausible mechanistic control on spatial patterns of Holocene temperature changes over this region. We suggest that cold conditions in the interior of the mid- to high-latitude Eurasian continent, in contrast to the

higher temperatures in surrounding regions, during the early to mid-Holocene, shaped the spatial patterns of long-term Holocene temperature changes over the mid-latitude Eurasian continent. Meanwhile, we do not deny the influence of insolation and greenhouse gas concentration on Holocene temperature changes, which could play decisive roles in temperature changes over the extratropical Eurasia continent during the mid- to late Holocene. Also, regions outside the influence of cold air masses during the early to mid-Holocene, e.g., Lake Yihsariwusu, should have followed the decreasing boreal insolation, displaying Holocene long-term cooling trends.

We have modified the text in this section (Line 244-248, 281-284) to make our statement clearer.

L196: "...might be broadly comparable to recent Arctic amplification and Eurasian cold winters". This does not seem to me a good analogy. Recent arctic amplification is thought to be caused by the rising CO₂, and the warming tends to occur uniformly. The Eurasian cooling is also for winter, opposite to the warm season proxy here. How are these consistent?

We understand that the forcings of Arctic warming during current winters and the early to mid-Holocene appear to be different. However, the spatial pattern of temperature conditions during the early to mid-Holocene, i.e., overall warm conditions in the Arctic Ocean and low temperatures over the central Eurasian continent, broadly resemble current winter conditions. We consider that the atmospheric circulation pattern during these two periods appears to be comparable, based on the resemblances in the spatial patterns of temperatures over the mid-latitude Eurasian during current winters and the early to mid-Holocene.

We also realized that using "Arctic amplification" might cause ambiguity, so we have modified this sentence to make our statement clearer (Line 276).

203: "As ice sheet-induced anticyclone might have persisted over northern mid-high latitudes (Fig. 4d), we suggest that large-scale changes in land-sea temperature and pressure gradients could induce more meridional, or wavy patterns of westerlies, and southward deflection of the polar front to continental Eurasia". What cause this southward shift, orbital, which season? CO₂? Ice sheet?

Here we meant that cold (sub)polar airmass reached the interior of the Eurasian continent due to the wavy patterns of westerlies. We have modified this sentence to clarify our statement (Line 287-288).

Should Ref 7 be Erb et al., 2022?

This ref. 7 (Osama) is for Last Glacial Maximum Reanalysis, and shows a warming trend in the Holocene. Erb et al. is on the Holocene with multiple models and does not show warming trend.

Erb, M. P. et al. Reconstructing Holocene temperatures in time and space using paleoclimate data assimilation. *Clim. Past* 18, 2599–2629 (2022).

Kaufman and Broadman (2023) show blended proxy data and simulated temperature at 6 ka relative to 0.5 ka using data from LGMR data assimilation by Osman et al. (2021), which indicates higher annual mean temperature over the extratropical Eurasian continent at 6 ka relative to 0.5 ka. We have removed the citation of Osman et al. (2021) from Line 60, 206, remained the citation of Kaufman and Broadman (2023), and added Erb et al. (2022) to our reference list.

Fig.2 and 3, each curve, include the lake name for each label, e.g. a) Lake.....b)Lake.....This helps readers...not have to go back to the caption.

We thank for this suggestion and have added site names to revised Fig. 2 and 3.

There is no Supplementary Table 2, referred in the caption of Fig.3. There is only supplementary data set 2. But that is not what a reviewer would like to see. A reviewer just needs the type, location et al of each record used.

We have modified filenames of Supplementary Tables.

Thanks again for carefully reading our manuscript, figures, and captions.

Reviewer #2 (Remarks to the Author):

In this manuscript, the authors present alkenone based temperature reconstructions for the Holocene: one from the monsoonal marginal zone and three from the continental/westerlies realm. By contextualization of the data they shape out two different spatial patterns of temperature changes throughout the study area and covered time period.

The overall aim of the manuscript is to solve question connected to the „Holocene conundrum“. A recent study (Herzschuh et al 2023; <https://cp.copernicus.org/articles/19/1481/2023/>) highlighted the importancy of focussing on spatial Holocene temperature patterns and their regional drivers, for the understanding of this proxy-model mismatch. By further investigating the situation in Asia, this manuscript by Jiang et al, is an important and interesting contribution to the debate.

Despite these aspects, I´m not convinced that this study is suitable for NCOMM for the following reasons:

1. I miss a more critical discussion of the alkenone proxy which is often problematic in lakes. I´m not familiar with all of the cited original studies including those of which data from other lakes were plotted, but it appears that many factors can influence the different kinds of uk37-proxies. Salinity changes, translocation of alkenone producers, glacial meltwater influence, community changes of alkenone producers,... are just some of the factors which

are mentioned in some of the cited papers. Partially, this is solved in the manuscript by comparison with independent proxies, but I still miss a discussion on how reliable alkenone-based temperature proxies are in the specific case, not only in the methods but in the main part of the manuscript.

We would like to clarify that we put relevant discussion in Methods previously. Meanwhile, we understand your concern and concur with you that alkenone proxies, like many other paleoclimatic indicators, are affected by various factors. It is important to consider the effects of different influencing factors when interpreting paleoclimatic records. We believe that the alkenone-based temperature records used in this study are reliable terrestrial temperature indicators in mid-latitude Asia for the following reasons:

- (1) The applicability of $U_{37}^{K'}$ index as a temperature indicator in mid-latitude Asian lakes has been examined through both regional calibrations and paleoclimate reconstructions. The $U_{37}^{K'}$ -temperature calibration based on lake surface sediment samples from mid-latitude Asia (Chu et al., 2005) is largely consistent with a culture experiment conducted in this region (Sun et al., 2007), indicating that $U_{37}^{K'}$ index could record the growing season temperature of alkenone-producing haptophytes in mid-latitude Asian lakes. Meanwhile, the $U_{37}^{K'}$ index has been successfully employed to reconstruct Holocene temperature changes over this region, after careful consideration of its suitability for different lakes (e.g., He et al., 2013; Jiang et al., 2022; Zhao et al., 2013).
- (2) Alkenone C_{37} isomer has not been identified from all four investigated mid-latitude Asian lakes, indicating that alkenones in these lakes are produced only by Group II haptophytes during the Holocene (Longo et al., 2013; Yao et al., 2022). In the absence of Group I haptophytes, the alkenone-based records in our study are not affected by phylogenetic effect. We have added an explanation of phylogenetic effect to revised main text (Line 87-92).
- (3) The similar variation trends of alkenone records from three Siberian lakes, combined with spatial patterns of $U_{37}^{K'}$ and $\%C_{37:4}$ records observed in investigated mid-latitude Asian lakes, led us to believe that variations in alkenone-based records in our study preliminary response to climatic changes rather than in-lake factors.
- (4) Alkenone $\%C_{37:4}$ records from investigated lakes could successfully indicate lake salinity variations and thus hydroclimatic changes during the Holocene. The $U_{37}^{K'}$ records from these lakes, following the structure of $\%C_{37:4}$ variations, could be largely confounded by salinity changes, while lake salinity has little effect on the $U_{37}^{K'}$ index. Therefore, we use $U_{37}^{K'}$ index as a reliable temperature indicator in our study.

To incorporate Reviewer's comment, we have moved the alkenone interpretation from the previous Methods section to the main text and also modified the text in this section (Line 81-125), which could provide a fair assessment of the reliability of alkenone records used in this study.

2. Connected with these aspects, I find the recent CP-paper by Herzschuh et al is more convincing, because solely focussing on pollen transfer functions, which have been proven to work well in context of precipitation and temperature reconstructions. Unfortunately, this

also takes away a bit of novelty from this new manuscript by Jiang et al, especially since Herzschuh et al already showed some spatial heterogeneity of Holocene temperature in Asia.

Concluding, I think this new manuscript is definitely be worth to be published after revisions, but more likely in a journal such as QSR or CP or maybe COMM EARTH ENV.

We did not cite Herzschuh et al. (2023) in our original manuscript, as it was formally published after our submission. However, we did cite Zhang et al. (2022), which also reports spatial heterogeneity in this region. We have now cited this recent study in our revised manuscript (Ref. 13 in revised manuscript), as it provides valuable insights into regional temperature changes and offers a significant pollen database for paleoclimatic studies. While we acknowledge the importance of pollen-based temperature reconstructions in paleoclimatic studies, we strongly believe that combining records inferred from different proxies could enhance our better understanding of the Holocene temperature changes, particularly regarding the suggested seasonality bias in temperature proxies. Meanwhile, we would like to clarify that the spatial patterns of Holocene temperature changes over mid-latitude Asia proposed in our study differ substantially from the latitudinal patterns discussed in Zhang et al. (2022) cited in our original manuscript and Herzschuh et al. (2023).

The study conducted by Herzschuh et al. (2023) investigated pollen-based latitudinal patterns of Holocene temperature changes from the Northern Hemisphere. Additionally, a recent synthesis of pollen records by Zhang et al. (2022), as cited in our original manuscript, also discussed the latitudinal pattern of Holocene temperature changes over the Northern Hemisphere. Pollen-based annual temperature records from Herzschuh et al. (2023) show long-term Holocene warming trends in 40-50°N of Asia, while cooling trends are observed in 60-70°N of Asia. Yet, pollen-based seasonal temperature records from Zhang et al. (2022) reveal Holocene long-term cooling trends in 0-50°N of Asia but warming trends in 50-75°N of Asia in both summer and winter seasons. Also, in Herzschuh et al. (2023) seasonal temperature trends could differ from annual trends, while in Zhang et al. (2022) seasonal trends are essentially the same as annual ones. Different pollen-based spatial patterns of Holocene temperature changes over mid-latitude Asia proposed by Zhang et al. (2022) and Herzschuh et al. (2023) indicate that Holocene temperature changes over this region may not conform to a latitudinal pattern. Therefore, conducting further studies that integrate temperature records inferred from various proxies and consider spatial patterns beyond latitudinal binning may provide new insights into Holocene temperature changes. We inset here (below) the figures by Herzschuh et al. (2023) and Zhang et al. (2022) to show different pollen-based latitudinal patterns of Holocene temperature changes over mid-latitude Asia.

Herzschuh et al. (2023)

Zhang et al. (2022)

Our alkenone records from mid-latitude Asian lakes, together with published temperature records inferred from different proxies, show that colder airmass appears to have prevailed in the interior of the mid-latitude Eurasian continent during the early to mid-Holocene, and the boundary between such distinct temperature patterns appears to be roughly situated along northern Xinjiang and may extend to the northeastern Tibetan Plateau, a spatial feature that has been largely overlooked in previous synthesis studies. We believe that the spatial patterns proposed in our study differ from the regional temperature synthesis based on latitudinal binning. Furthermore, our results may contribute to a better understanding of the inconsistent pollen-based latitudinal patterns proposed by Zhang et al. (2022) and Herzschuh et al. (2023).

In addition, there has been a longstanding discussion regarding potential seasonality bias in proxy-based temperature reconstructions. By comparing our alkenone results with other records inferred from various proxies potentially biased toward different seasons, we suggest that the spatial patterns of Holocene temperature changes over mid-latitude Asia are unlikely to be attributed to seasonality bias in proxies. We understand the Reviewer's concern that comparing records inferred from different proxies may increase uncertainty, so we focused on comparing the long-term Holocene temperature trends rather than the quantitative changes in

temperature anomalies. Starting with alkenone records, we identified different temperature variation trends from mid-latitude Asian lakes. We then compared the spatial patterns inferred from alkenone results with other temperature records inferred from various proxies, models, and data assimilations, and proposed that colder air mass appears to have prevailed in the interior of the mid-latitude Eurasian continent during the early to mid-Holocene. Such spatial pattern is unlikely to be attributed to seasonality bias in proxies and is independent of latitude positions.

Overall, we are confident that our study presents sufficient novelty. The spatial patterns we suggest, which differ from pollen-based latitudinal patterns, might account for the incoherent pollen-based spatial patterns proposed by Zhang et al. (2022) and Herzschuh et al. (2023). Our study challenges the proposed seasonality bias in proxies and modeled spatial patterns, highlighting that spatial patterns of Holocene temperature changes should be re-considered in both record integrations and model simulations, with important implications for terrestrial hydroclimate changes. Our results also challenge current approaches addressing the Holocene temperature conundrum (at least for this region), implying that several current views concerning the Holocene temperature conundrum need to be re-considered.

To address Reviewer's comment, we have included Herzschuh et al. (2023) in the revised main text (Line 63-67) and reference list (Ref. 13 in revised manuscript). We have also emphasized that the spatial patterns proposed in our study differ from the pollen-based latitudinal patterns, (revised Fig. 5, Line 162-164, 175-176, 194-196).

Minor comments:

- The above mentioned paper by Herzschuh needs to be cited

We have included Herzschuh et al. (2023) in reference list (Ref. 13 in revised manuscript).

- Maps in Fig 3a and Ext Fig 5a would benefit from labelling the dots (as in Fig. 1)

We appreciate the suggestion. It is worth noting that previous Fig. 3a (revised Fig. 4a) contain ~160 records. Labeling each dot would make the figures complicated and challenging to read. Our primary objective is to highlight the spatial patterns of temperature changes over mid-latitude Eurasia in these figures. Actually, we have provided a list of records presented in both figures in Supplementary Table 2, which includes detailed site locations and references. To incorporate Reviewer's comment while maintaining the readability of the figures, we have added site location names to revised caption of Fig. 1, and added site numbers same as Fig. 1 to revised Fig. 4a.

- There is a lot of reference to figures in the appendix. I would recommend to reconsider if some of those figures could be shifted to the main manuscript, in case this manuscript is re-arranged to a less short-format version.

We very much appreciate this suggestion. We have moved the alkenone interpretation section from Methods to revised main text (Line 81-125), and now most of references cited in this study have been included in revised main text. We have modified and moved two supplementary figures to the main manuscript (revised Fig. 4, 5) to better address our statement.

References

1. Alder, J. R. & Hostetler, S. W. Global climate simulations at 3000-year intervals for the last 21000 years with the GENMOM coupled atmosphere–ocean model. *Clim. Past* **11**, 449–471 (2015).
2. Bader, J. et al. Global temperature modes shed light on the Holocene temperature conundrum. *Nat. Commun.* **11**, 1–8 (2020).
3. Brierley, C. M. et al. Large-scale features and evaluation of the PMIP4-CMIP6 mid Holocene simulations. *Clim. Past* **16**, 1847–1872 (2020).
4. Chu, G. et al. Long-chain alkenone distributions and temperature dependence in lacustrine surface sediments from China. *Geochim. Cosmochim. Acta* **69**, 4985–5003 (2005).
5. Erb, M. P. et al. Reconstructing Holocene temperatures in time and space using paleoclimate data assimilation. *Clim. Past* **18**, 2599–2629 (2022).
6. Harrison, S. P., Prentice, I. C., & Bartlein, P. J. Influence of insolation and glaciation on atmospheric circulation in the North Atlantic sector: implications of general circulation model experiments for the Late Quaternary climatology of Europe. *Quat. Sci. Rev.* **11**, 283–299 (1992).
7. He, Y. et al. Late Holocene coupled moisture and temperature changes on the northern Tibetan Plateau. *Quat. Sci. Rev.* **80**, 47–57 (2013).
8. Herzsuh, U. et al. Regional pollen-based Holocene temperature and precipitation patterns depart from the Northern Hemisphere mean trends. *Clim. Past* **19**, 1481–1506 (2023).
9. Jiang, Q. et al. Exceptional terrestrial warmth around 4200–2800 years ago in Northwest China. *Sci. Bull.* **67**, 427–436 (2022).
10. Kaufman, D. S. & Broadman, E. Revisiting the Holocene global temperature conundrum. *Nature* **614**, 425–435 (2023).
11. Liu, Z. et al. Transient simulation of last deglaciation with a new mechanism for Bølling-Allerød warming. *Science* **325**, 310–314 (2009).
12. Longo, W. M., Dillon, J. T., Tarozo, R., Salacup, J. M. & Huang, Y. Unprecedented separation of long chain alkenones from gas chromatography with a poly (trifluoropropylmethylsiloxane) stationary. *Org. Geochem.* **65**, 94–102 (2013).
13. Osman, M. B. et al. Globally resolved surface temperatures since the Last Glacial Maximum. *Nature* **599**, 239–244 (2021).
14. Siegert, M. J. & Dowdeswell, J. A. Numerical reconstructions of the Eurasian Ice Sheet and climate during the Late Weichselian. *Quat. Sci. Rev.* **23**, 1273–1283 (2004).
15. Sun, Q., Chu, G., Liu, G., Li, S. & Wang, X. Calibration of alkenone unsaturation index with growth temperature for a lacustrine species, *Chrysotila lamellosa* (Haptophyceae). *Org. Geochem.* **38**, 1226–1234 (2007).
16. Wassenburg, J. A. et al. Reorganization of the North Atlantic Oscillation during early Holocene deglaciation. *Nat. Geosci.* **9**, 602–605 (2016).

17. Yao, Y. et al. Phylogeny, alkenone profiles and ecology of Isochrysidales subclades in saline lakes: Implications for paleosalinity and paleotemperature reconstructions. *Geochim. Cosmochim. Acta* **317**, 472–487 (2022).
18. Zhao, C. et al. Holocene temperature fluctuations in the northern Tibetan Plateau. *Quat. Res.* **80**, 55–65 (2013).
19. Zhang, W. et al. Holocene seasonal temperature evolution and spatial variability over the Northern Hemisphere landmass. *Nat. Commun.* **13**, 5334 (2022).

Reviewer #1 (Remarks to the Author):

The authors have addressed most of my questions reasonably well. Therefore, I recommend publication. However, there are two comments that they have to clarify before the acceptance.

1. Since seasonality is a crucial factor here, it is important to specify the warming for annual mean or some season. For example, in the abstract, for line 27, 28, 29,30, warming or cooling are used, but without specification of its seasonality/annual mean. This should be corrected! Also in the text.

2. Fig.4b, c, d: Is it the wrong sign? The caption says it is the trend from 6ka to 0ka. But I think this is the wrong sign. Please check!

As a check, and also useful for readers, the authors can also plot on Fig.6e, f, the annual mean trend of model, say, in a dotted line.

Reviewer #2 (Remarks to the Author):

This is the revised version of a manuscript submitted to NCOMMS by Jiang and co-authors.

A notify and appreciate that the authors extensively responded to all of the reviewers comments and made substantial changes that lead to significant improvement of the manuscript. The critical discussion about value of alkenone based temperature proxies and potential bias by in-lake effects was shifted to the main part of the manuscript, which I consider as important, given the complications in proxy interpretation that often arise due to those effects. Further, discussion on potential seasonality effects of different proxy groups comes out more clearly in the revised version. Finally, the model-proxy comparison aspect was much strengthened in the new version, which significantly improves the discussion on spatial trends in temperature change in the studied region.

Some scepticism about bias on alkenone based temperature proxies remain. However, in context of the compiled data, the model-proxy comparison, and the arguments given by the authors, the presented data, their contextualisation, and inferred conclusions contain sufficient novelty and quality to push the debate about the Holocene conundrum forward.

I therefore recommend this manuscript for publication in NCOMMS.

We have carefully followed Editor's and reviewers' comments revising our manuscript. In order to clearly highlight the changes made in the revision, we have copied all these comments below in black and inserted our replies in blue. Line numbers mentioned in our response letter refer to the ones in our revised manuscript.

Reviewer #1 (Remarks to the Author):

The authors have addressed most of my questions reasonably well. Therefore, I recommend publication. However, there are two comments that they have to clarify before the acceptance.

1. Since seasonality is a crucial factor here, it is important to specify the warming for annual mean or some season. For example, in the abstract, for line 27, 28, 29,30, warming or cooling are used, but without specification of its seasonality/annual mean. This should be corrected! Also in the text.

We used “long-term cooling trend in warm season temperatures” in Line 27 to specify the seasonality of alkenone records. Existing temperature records from surrounding regions, in Line 29, refer to records inferred from different proxies which might indicate temperature changes in different seasons, so we do not point out the seasonality here (although they are specified in the main text). We have also checked our text to specify the seasonality. Meanwhile, we understand that defining the seasonality of certain temperature reconstructions based on multiproxy data can be challenging, while the seasonality of temperatures from model simulations is clearly stated. Therefore, given these circumstances, we would like to retain the original description in Abstract as it stands.

2. Fig.4b, c, d: Is it the wrong sign? The caption says it is the trend from 6ka to 0ka. But I think this is the wrong sign. Please check!

We appreciate this comment and have modified signs in Fig. 4b, c, d.

As a check, and also useful for readers, the authors can also plot on Fig.6e, f, the annual mean trend of model, say, in a dotted line.

We have added annual mean temperature trends to Fig. 6e, f.

Reviewer #2 (Remarks to the Author):

This is the revised version of a manuscript submitted to NCOMMS by Jiang and co-authors.

A notify and appreciate that the authors extensively responded to all of the reviewers comments and made substantial changes that lead to significant improvement of the manuscript. The critical discussion about value of alkenone based temperature proxies and potential bias by in-lake effects was shifted to the main part of the manuscript, which I consider as important, given

the complications in proxy interpretation that often arise due to those effects. Further, discussion on potential seasonality effects of different proxy groups comes out more clearly in the revised version. Finally, the model-proxy comparison aspect was much strengthened in the new version, which significantly improves the discussion on spatial trends in temperature change in the studied region.

Some scepticism about bias on alkenone based temperature proxies remain. However, in context of the compiled data, the model-proxy comparison, and the arguments given by the authors, the presented data, their contextualisation, and inferred conclusions contain sufficient novelty and quality to push the debate about the Holocene conundrum forward.

I therefore recommend this manuscript for publication in NCOMMS.

We greatly appreciate Reviewer's positive comments.